

# A coupled atmospheric-hydrologic modeling system with variable grid sizes for rainfall-runoff simulation in semi-humid and semi-arid watersheds: How does the coupling scale affects the results?

Jiyang Tian[1], Jia Liu[1], Yang Wang[1,2], Wei Wang[1,3], Chuanzhe Li[1], Chunqi Hu[4]

5  [1] State Key Laboratory of Simulation and Regulation of Water Cycle in River Basin, China Institute of Water Resources and Hydropower Research, Beijing, 100038, China
[2] Swanson School of Engineering, University of Pittsburgh, Pittsburgh, PA15261, USA
[3] College of Hydrology and Water Resources, Hohai University, Nanjing, 210098, China
[4] Bureau of Water Resources Survey of Heibei, Shijiazhuang, 050031, China

10  *Correspondence to*: Jia Liu (hettyliu@126.com)

**Abstract.** The coupled atmospheric-hydrologic modeling system is an effective way in improving the accuracy of rainfall-runoff modeling and extending the lead time in real-time flood forecasting. The aim of this study is to explore the appropriate coupling scale of the coupled atmospheric-hydrologic modeling system, which is established by the Weather Research and Forecasting (WRF) model and the gridded Hebei model with three different sizes (1×1km, 3×3km and 9×9km). The soil moisture storage capacity and infiltration capacity of different grids in the gridded Hebei model are obtained and dispersed using the topographic index. The lumped Hebei model is also used to establish the lumped atmospheric-hydrologic coupled system as a reference system. Four 24 h storm events occurring at two small and medium-scale sub-watersheds in northern China are selected as cases study. Contrastive analyses of the flood process simulations from the gridded and lumped systems are carried out. The results show that the flood simulation results may not always be improved with higher dimension precision and more complicated system, and the grid size selection has a great relationship with the rainfall evenness. For the storm events with uniform spatial distribution, the coupling scale has less impact on flood simulation results, and the lumped system also performs well. For the storm events with uneven spatiotemporal distribution, the corrected rainfall can improve the simulation results significantly, and higher resolution lead to better flood process simulation. The results can help to establish the appropriate coupled atmospheric-hydrologic modeling system to improve the flood forecasting accuracy.



## 1 Introduction

Compared to the traditional flood forecast models that take gauge precipitation as inputs, the coupling of a mesoscale Numerical Weather Prediction (NWP) model with a hydrological model has been proven to be an effective way in improving the accuracy of rainfall-runoff modeling and extending the lead time in real-time flood forecasting (Wu et al, 2014; Wan and

Xu, 2011). Atmosphere-hydrologic coupled system becomes world current technology frontier for flood forecasting and correlation studies have received extensive attention (Robert et al, 2000; Tian et al, 2019). Given the multi-scale nature of watershed hydrologic models and rainfall data output by numerical atmospheric models, one key to obtaining accurate simulations results is the establishment of a suitable atmosphere-hydrologic coupled system.

Most atmosphere-hydrologic coupling studies tend to use lumped or distributed hydrological models (constructed in research

watersheds) to run atmospheric models based on hydrological model requirements by inputting meteorological information such as rainfall (Wagner et al, 2016). Liu et al. (2015) simulated the floods by a lumped hydrological model integrated with the weather research and forecasting (WRF) model in 10km scale at a small catchment (135.2km$^2$). Rogelis et al. (2018) focused on comparison of streamflows obtained from three different lumped hydrological models driven by WRF precipitation forecast at Tunjuelo River basin (380km$^2$). The spatial resolution of 1.67km was set for finest domain in WRF model. Li et al.

(2017) extended the flood forecasting lead time in Liujiang River basin (58270km$^2$) by WRF precipitation forecast with a distributed hydrological model. The WRF precipitation forecast with the resolution being 20km × 20km was downscaled to 200m × 200m, which is suite for the resolution of the hydrological model. However, most studies have no consideration for the spatial-scale matching issue between atmospheric and hydrological models,which has significant impact on the flood forecast accuracy.

Model complexity is another impact factor that must be considered during model construction. When investigating the complexity of data-driven models, Myung et al. (2009) found that the relationship between model complexity and error can be described as Fig. 1. It is outlined that model selection/construction should be based not solely on the goodness-of-fit, but must also consider the model complexity (whether the model's description of observed data is achieved in the simplest possible manner). This is because a highly complex model can provide a good fit without necessarily bearing any interpretable

relationship with the underlying process. It is shown that model selection based solely on the fit to observed data will result in the choice of an unnecessarily complex model that overfits the data, and thus generalizes poorly (Myung, 2000).

[Figure 1]

This could also be true when dealing with the coupled atmospheric-hydrologic systems. Although the models in the coupled system are physically-based with deterministic structures and parameters, the complexity is to a great extend decided by

choosing an appropriate spatial scale (grid size) for coupling (Verbunt et al, 2006; Hostache et al, 2011; Rogelis and Micha, 2018). A smaller coupling scale between the atmospheric and the hydrologic models helps utilize higher-resolution rainfall information, but a higher grid resolution results in a greater computation load, leading in turn to greater cumulative errors in





the simulated discharge at the watershed outlet. On the other hand, a larger coupling scale allows a smaller computation load, but the accuracy of rainfall information decreases such that the final simulation results may be subject to a higher error rate. Therefore, clarifying the variation pattern of the coupled system at different spatial scales can benefit research on the atmospheric-hydrologic coupling and improve the accuracy of rainfall-runoff simulation and forecasting. Within this scope,

important issues that should be addressed in depth include how to effectively use the high-resolution rainfall information provided by atmospheric models, how to select a suitable spatial scale for constructing a coupled atmospheric-hydrologic system, and how to make tradeoffs between the computational efficiency and accuracy. Moreover, to date, the spatial scale of atmospheric-hydrologic coupling has not yet been investigated with respect to different rainfall processes.

In this study, a coupled atmospheric-hydrologic system with variable grid scales is firstly developed based on the construction

of a gridded hydrologic model in combination with a mesoscale NWP model to provide rainfall driving data at different resolutions; this allows the atmospheric-hydrologic coupling at different spatial scales. The most recently-developed mesoscale NWP model, i.e. Weather Research & Forecasting (WRF) model, is adopted to provide the downscaled rainfall information and the Hebei model, a traditional model with mixed flow mechanisms of both saturation-excess and infiltration-excess, is used to construct the gridded hydrologic model. Three different coupling scales are considered in this study, which is 1×1 km,

3×3 km, and 9×9 km. Next, the coupled atmospheric-hydrologic system with different spatial scales to reflect the degree of model complexity, is applied to four storm events in semi-humid and semi-arid watersheds of Northern China. The storm events are characterized by different representatives of the rainfall distribution evenness in space and time. The performance of the coupled system in different degree of complexity (i.e., with different coupling scale or grid size) is fully investigated with regard to different storm events, and the relationship between the rainfall evenness and the selection of the most

appropriate coupling scale is further discussed.

## 2 Data sources

### 2.1 Storm events

Based on the analyses of the historical floods in the study area, four representative 24h-duration storm events with relatively high flow peaks (Fig. 2) were selected and used to test the performance of the coupled atmospheric-hydrologic system for

rainfall-runoff modeling. The start and end times together with the 24h accumulative rainfall amounts and the peak discharges are listed in Table 1. Among the four events, three occurred in Fuping sub-watershed and one occurred in Zijingguan sub-watershed.

[Figure 2 and Table 1]

The coefficient of variation $C_v$ was used as an indicator to verify the representativeness of the selected storm events and to

characterize their temporal and spatial distributions. With regard to the spatial distribution, the coefficient reflects the variations of the 24h cumulative rainfall among different rainfall stations across the watershed. As for the temporal distribution, it shows





the variations of the average areal rainfall at different time steps during the whole storm event. The coefficient of variation $C_v$ can thus be calculated as:

$$C_v = \sqrt{\frac{\sum_{i=1}^{n}(\frac{x_i}{\bar{x}}-1)^2}{n}} \qquad (1)$$

For spatial distribution calculations, $x_i$ is the 24h cumulative rainfall at a certain rainfall station $i$, $\bar{x}$ represents the average
cumulative rainfall of all stations, and $n$ refers to the number of rainfall stations in the sub-watershed. For temporal distribution calculations, $x_i$ is the areal rainfall at the $i$-th time step, $\bar{x}$ is the average of the areal rainfall of all time steps, and $n$ refers to total time steps the rainfall lasts for, which is 24 hours in this study. The calculated spatial and temporal values for the four selected events can be found in Table 2.

**[Table 2]**

Tian et al. (2017a) stated that the northern China has experienced very few rainfall events with both even distributions in space and time. Based on the historical storm events in the study area, and using 5% as a cutoff, the critical $C_v$ value would then be 0.40 for even spatial distribution of rainfall and 1.00 for even temporal distribution (Tian et al., 2017a). In this case, among the four storm events selected in this study, event 1 has evenly distributed rainfall in both space and time, and the rainfall of event 2 is evenly distributed in space but unevenly distributed in time. Both event 3 and event 4 have unevenly distributed
rainfall in space and time, while event 4 which can be noted with much larger $C_v$ values and higher accumulative rainfall is considered to be an extreme case. The storm happened in a very short period of time and in a very concentrated area with high rainfall intensity.

**2.2 Study site**

In this study, two small and medium-scale sub-watersheds of the Daqing River watershed (the Fuping sub-watershed of the
south branch and the Zijingguan sub-watershed of the north branch) were selected in this study (Fig. 3 and Fig. 4). The Fuping sub-watershed has a total area of 2,219 km$^2$ and is located in the upper reaches of the Zhishahe River, a south branch of the Daqing River. The Zijingguan sub-watershed has a total area of 1,760 km$^2$ and is located in the upper reaches of the Juma River, a north branch of the Daqing River. These two mountainous sub-watersheds collectively represent the rainfall-runoff characteristics of the sub-humid and sub-arid area in Northern China. Rainfall in the Daqing River watershed is characterized
by events with a short duration and a large intensity, which happens in mountainous area such as the two study sub-watersheds is more likely to result in severe flood disasters.



## 3. Experimental design with WRF and the grid-based hydrological model

### 3.1 Atmospheric modeling

#### 3.1.1 WRF model setup

The WRF model with a variety of physical and numerical options has widely been used for numerical rainfall prediction in the
catchment scale (Liu et al., 2012; Cassola et al., 2015; Tian et al., 2017b). In this study, three nested domains were adopted,
centering over the Fuping sub-watershed (at 39°04′15″N, 113°59′26″E) or the Zijingguan sub-watershed (39°25′59″N,
114°46′01″E). The two sub-watersheds were totally covered by the innermost domain with the interaction between the parent
and child domain being allowed. The horizontal grid spacing was set to 1 km, 3 km and 9 km from the innermost to the
outermost domain, which was in consistence with the resolution of the hydrological model and allowed for easy operation
during coupling (Tian et al., 2017a). The NCEP/NCAR final operational global analysis (FNL) data with a precision of 1°×1°
were used to provide the lateral and boundary conditions of the WRF model (Wang et al., 2013a,b). The integration time step
of the model was set to 6 s and the output data interval was set to 1 h (Hong and Lee, 2009). Forty layers are considered in the
three nested domains with a top-layer pressure of 50 hPa. The projection method was chosen to be Lambert, which has good
applicability in the mid-latitude regions. The other basic settings of the WRF model adopted in this study can be found in Table
15  3.

**[Table 3]**

#### 3.1.2 Physical parameterisations

The WRF rainfall simulation is sensitive to the selection and the combination of its physical parameterisations (Di et al., 2015).
In this study, the well-performed and extensively-used parameterisations in Northern China were chosen (Miao et al., 2011;
Di et al., 2015, Tian et al., 2017a), which include two microphysics parameterisations, i.e, Lin (Lin et al., 1983) and WSM6
(Hong et al., 2006), two cumulus parameterisations, i.e., KF (Kain, 2004) and GD (Grell and Freitas, 2014), and two PBL
(planetary boundary layer) parameterisations, i.e., MYJ (Hong et al., 2006) and YSU (Janjić, 1994). Besides, RRTM and
Dudhia (Evans et al., 2012) usually cooperated well as the long/short wave radiation parameterisations and Noah was chosen
to be the land surface model (Chen et al., 2014). Table 4 shows the key parameterisations chosen for the four storm events in
this study, which were based on their performances on the simulation of different rainfall characteristics.

**[Table 4]**

### 3.2 Hydrologic modeling

#### 3.2.1 Construction of a grid-based hydrologic model

A mesoscale NWP model can produce downscaled rainfall data with different horizontal resolutions through the multilayer
nested grids. In order to use the spatial information of the WRF rainfall output data, a loosely structured hydrological model



was constructed in each of the two sub-watersheds. The sub-watersheds were firstly divided into grids, based on which a rainfall-runoff model was established on each grid cell to calculate the runoff generation and concentration within the grid cell. After that a spatially distributed river network model was used to calculate the confluence of the river flow from the grids to obtain the total discharge at the outlet of the watershed. As the scale of grid-based hydrologic models is dependent on the grid

size, it is possible to conduct the grid division of the watershed according to the horizontal resolution of the WRF output rainfall data.

In this study, the 30 m digital elevation model (DEM) provided by Geospatial Data Cloud (http://www.gscloud.cn/) assisted in establishing the grid-based hydrologic model. Based on the DEMs of the two sub-watersheds and the GIS spatial analysis tools, the flow direction of each grid cell was determined, the river network was generated and the computing orders of the

grids were then obtained. Considering the horizontal resolution of the WRF model in this study was set to be 1 km, 3 km and 9 km from the innermost to the outermost domain, the two sub-watersheds, Fuping and Zijingguan, were divided into grids with the same three resolutions. The divided 1×1 km, 3×3 km, and 9×9 km grids of the two sub-watersheds with a distinction between channel and non-channel cells are shown in Fig. 5 and Fig. 6.

**[Figure 5 and 6]**

**3.2.2 Runoff generation and water exchange processes**

In this study, a conceptual rainfall-runoff model, named Hebei model, was built on each grid cell of the two sub-watersheds. The Hebei model was specially developed for rainfall-runoff modeling in the semi-humid and semi-dry area of Northern China, and has widely been applied in Hebei Province by considering both infiltration-excess and saturation-excess mechanisms of the runoff generation. The structure of the Hebei rainfall-runoff model is shown in Fig. 7. $\gamma$ is the proportion of the area with

the infiltration capability beyond a certain value, $\chi$ is the proportion of runoff generation area to entire basin area, $W$ is the storage capacity of the point in the basin, and $WMM$ is the storage capacity of the maximum point in the basin. The surface runoff is generated when the rainfall intensity exceeds the infiltration capacity, and when the infiltrated water volume matches the soil's water shortage capacity, the surplus water generates the groundwater runoff. The total flow at the watershed outlet is composed of both the surface and the underground runoff.

**[Figure 7]**

The infiltration curve of the model can be expressed as:

$$f = \left( i - \frac{i^{(1+n)}}{(1+n)f_m^n} \right) e^{-um} + f_c \tag{2}$$

where $f$ is infiltration rate (mm/h), $i$ is rainfall intensity (mm/h), $n$ is an exponent, $f_c$ is the stable infiltration rate (mm/h), $m$ is the surface soil moisture (mm), $u$ is an exponent indicating the decreasing speed of the infiltration rate with the increase of the

soil moisture, and $f_m$ is the infiltration capacity inside the grid (mm/h).

When the calculation period is $t$ hours, the infiltration volume in that time period can be calculated as:



$$F_t = \sum_{i=1}^{t} \overline{f_i} \tag{3}$$

where $\overline{f_i}$ is the accumulation infiltration volume in $i$-th time period.

Thus, surface runoff $R_s$ in a given time period can be expressed as:

$$Rs = P_t - E_t - F_t \tag{4}$$

where $P_t$ is the precipitation volume (mm), $E_t$ is the evaporation volume (mm), and $F_t$ is the infiltration volume (mm) in the time period $t$. The calculation of the underground runoff ($R_g$) is the same with the Xin'anjiang model. Detailed descriptions are shown by Zhao (1992).

According to the principles of infiltration volume calculation in the time period and the spatial distribution of the water storage capacity in the grid, the groundwater runoff in the time period can be calculated as:

When $P_a + F_t < W_m^{'}$,

$$R_g = F_t - E_t + P_a - W_m + W_m(1 - \frac{F_t + P_a}{W_m^{'}})^{(1+b)} \tag{5}$$

When $P_a^{'} + F_t \geq W_m^{'}$,

$$R_g = F_t - E_t + P_a - W_m \tag{6}$$

where $R_g$ is the groundwater runoff in the time period (mm), $E_t$ is the evaporation volume in the time period, $P_a$ is the influenced

precipitation during the early phase of the grid (mm), $W_m$ is the average storage capacity of the grid (mm), $W_m^{'}$ is the storage capacity of the grid at the maximum point (in mm), and $b$ is the exponent of the water storage capacity curve.

**(1) Exchange of water in a non-channel grid cell**

The exchange of water between non-channel grid cells is illustrated in Fig. 8, where $P$ and $E$ is the rainfall and the evaporation in the grid cell, $Q_{Si}$ and $Q_{Gi}$ is the surface runoff inflow and the underground runoff inflow from the upstream grid cell, $Q_S$ and

$Q_G$ is the surface runoff and the groundwater runoff, and $Q_{So}$ and $Q_{Go}$ is the surface and the groundwater runoff outflow to the downstream grid cell.

**[Figure 8]**

As mentioned, when the rainfall intensity in the grid cell is greater than the infiltration capacity, the surface runoff $Q_S$ occurs in this cell. If the upstream cell also generates the surface runoff $Q_{Si}$, the surface runoff outflow $Q_{So}$ can be calculated as:

$$Q_{So} = Q_{Si} + Q_S \tag{7}$$

As part of the groundwater runoff from the upstream grid cell supplements the soil water storage, the ratio of which is set as $sc$, the groundwater runoff outflow $Q_{Go}$ is calculated as:



$$Q_{Go} = Q_{Gi} \times (1 - sc) + Q_G \tag{8}$$

The soil water storage of the grid cell $P_{at}$ at the end of the period $t$ is:

$$P_{a_t} = P_{a_{t-1}} + Q_{Gi} \times sc \tag{9}$$

If the soil water storage $P_{at}$ is greater than the storage capacity of the grid cell, then let $P_{at}$ equal to the soil moisture capacity.

5 **(2) Exchange of water in a channel grid cell**

For the exchange of water in a channel grid cell, the generated surface and groundwater runoff is no longer supplied directly to the downstream cell, but imported to the river channel and passes through the channel routing to the next cell. The processes are as illustrated in Fig. 9, the symbols of which have the same meanings as Fig. 8.

**[Figure 9]**

10 With the rainfall $P$ and the evaporation $E$, the surface runoff $Q_S$ and the groundwater runoff $Q_G$ are generated. Based on the flow routing, the surface and the groundwater inflow to the river channel $Q_{Sr}$ and $Q_{Gr}$ can be obtained. The generated total inflow $Q_c$ of the channel grid is then calculated as:

$$Q_c = (Q_{Sr} + Q_{Gr}) \times \varepsilon \tag{10}$$

where $\varepsilon$ is the proportional coefficient of the surface and the groundwater runoff into the river channel. By adding $Q_{in}$ and 15 $Q_c$, the outflow of the channel grid cell $Q_{out}$ to the downstream cell can be obtained as:

$$Q_{out} = Q_{in} + Q_c \tag{11}$$

**3.2.3 Confluence calculation and channel seepage**

The storage-discharge relation of grid cell can be simplified as a single line forme. The storage-discharge equation can be obtained as:

20 $$S = \frac{A}{1 - \omega} Q^{1 - \omega} \tag{12}$$

where $S$ is water storage (m³), $Q$ is the flow out rate (m³/s), A is the confluence parameter and $\omega$ is the shape parameter. According to the water balance equation:

$$\frac{dS}{dt} = I - Q \tag{13}$$

the flow out rate of the grid cell $Q_t$ at the end of the period $t$ is:





$$
Q_t = \begin{cases}
Q_{t-1}\left(1+\dfrac{\omega}{A}Q_{t-1}^{\omega}\right)^{-\frac{1}{\omega}} & I_{t-\tau}=0 \\[2ex]
\left[0.5\left(16A^2+8I_{t-\tau}+16AQ_{t-1}^{0.5}-4Q_{t-1}\right)^{0.5}-2A\right]^2 & \omega=0.5 \\[2ex]
e^{\left(\frac{I_{t-\tau}-0.5Q_{t-1}+S_{t-1}+A\ln Q_{t-1}-0.5Q_t}{A}\right)} & \omega=1 \\[2ex]
\left[\dfrac{I_{t-\tau}-0.5Q_{t-1}+DQ_{t-1}^{1-\omega}-0.5Q_t}{D}\right] & \omega\neq1
\end{cases}
\tag{14}
$$

where $I$ is the inflow rate (m³/s), $I_{t-\tau}$ is the inflow rate (m³/s) at time $t$-$\tau$ ($\tau$ is the travel time of flood wave) and $D$ can be expressed as:

$$
D=\frac{A}{(1-\omega)\Delta t}
\tag{15}
$$

where $\Delta t$ is the calculation time interval ($s$).

Due to the perennial water shortage and great channel seepage in study site, Horton infiltration model is applied to obtain the infiltration volume. When the calculation interval is one hour, the infiltration volume can be calculated by the following equation:

$$
f=(f_0-f_c)\bullet e^{-kt}+f_c
\tag{16}
$$

where $f_0$ is the initial infiltration rate (mm/h), $k$ is a parameter about soil. The channel seepage should be deducted before channel confluence is calculated with Eq. (14).

### 3.2.4 Discretization method for the soil moisture storage capacity and the infiltration capacity

In a grid-type hydrological model, it is necessary to determine the soil moisture storage capacity and infiltration capacity in the grid cells in order to calculate the runoff generated within the grid and to simulate the generation and convergence of runoff

in the river basin. Therefore, the determination of soil moisture storage capacity and infiltration capacity are key to model construction. Beven and Kirky (1979) proposed a topography-based semi-distributary hydrological model (TOPMODEL) that fully considered the effects of topography on the formation and change of runoff areas, using the spatial distribution of the terrain index $ln(\alpha/tan\beta)$ to reflect the spatial distribution of saturated and deficit water volumes in the river basin. Based on this theory, it can be assumed that areas with similar topographic indices have the same hydrological response.

According to a statistical analysis of topographic indices for the 1 × 1 km, 3 × 3 km, and 9 × 9 km grids in the Fuping and Zijinguan basins, the cumulative distribution curves of the different grids' topographic indices in the same area have the same shape (parabolic). However, the cumulative distribution curve of topographic indices between different areas is also very similar. Experimentation showed that the soil moisture storage capacity and infiltration capacity of different grids can be obtained and dispersed using the topographic index as follows:



$$\frac{W_i}{WMM} = exp\left\{-\left[\frac{\ln(TI_i - TI_{min} + 1)}{\alpha}\right]^{\beta}\right\} \tag{17}$$

where $W_i$ is the moisture storage capacity of a certain grid cell (mm), $TI_i$ is the topographic index of the grid, $TI_{min}$ is the topographic index of the minimum point in the basin, $\alpha$ is the scale parameter of the grid, and $\beta$ is the shape parameter of the grid.

$$\frac{f_i}{f_m} = \left\{1 - \left[1 - exp[-\frac{1}{\alpha}[\ln(T_i - TI_{min} + 1)]^{\beta}]^b\right]\right\}^{1/n} \tag{18}$$

where $f_i$ is the infiltration capacity of a certain grid cell in the river basin, and $f_m$ is the infiltration capacity of the maximum point in the basin.

### 3.3 Establishment of coupled atmospheric-hydrologic systems

Coupling can be achieved either one-way or two-way. For the latter, it is necessary to establish a communication mechanism between the atmospheric and hydrologic models to allow both to respond and combine dynamically. However, this process requires more complicated mechanisms and a higher amount of supportive data, making this approach difficult to use widely in actual forecasting, so this study focused on one-way coupling. In order to research coupling scales, it is necessary to establish a gridded atmospheric-hydrologic coupled system. The WRF model was used to generate gridded rainfall data and the gridded Hebei model was used as the land surface hydrologic model. The latter was set up with different grid divisions to allow the retrieval of rainfall data with a corresponding precision from the output data of the WRF model. Differences in the output processes of the hydrological models at different scales were then analyzed and the relationships between the differences and rainfall patterns were considered. The lumped Hebei model was also used to establish the lumped atmospheric-hydrologic coupled system. The gridded rainfall data from WRF model was averaged over each sub-watershed, which was regarded as the input of the lumped Hebei model. The coupled atmospheric-hydrologic system is illustrated in Fig. 10.

**[Figure 10]**

The gridded Hebei model was divided into the three different grid sizes noted above. Grid center coordinates could be used to retrieve the corresponding output data from the WRF model for driving the hydrologic model. For parameter calibration, seven flood processes in the Fuping sub-watershed and six in the Zijingguan sub-watershed from 1996–2013 were selected; another two were selected for model verification with an average Nash efficiency coefficient of up to 0.686, indicating that the established land-atmosphere coupled system generated good simulation results.

### 3.4 Evaluation statistics

The flood simulation results were evaluated using three statistics: the relative flood peak error ($R_f$), the relative flood volume error ($R_l$), and the Nash-Sutcliffe efficiency coefficient (NSE).





$$R_f = \left(Q_f{'} - Q_f\right) / Q_f \tag{19}$$

$$R_1 = \left(Q_1{'} - Q_l\right) / Q_l \tag{20}$$

$$NSE = 1 - \frac{\sum_{i=1}^{N}\left(Q_i{'} - Q_i\right)^2}{\sum_{i=1}^{N}\left(Q_i - \overline{Q}\right)^2} \tag{21}$$

where $Q_f{'}$ and $Q_f$ is the simulated and observed flood peak flow, $Q_l{'}$ and $Q_l$ is the simulated and the observed flood volume, $Q_i{'}$ and $Q_i$ is the simulated and observed flow discharge at the $i$-th time step, $N$ is the total time steps of the flood event, and $\overline{Q}$ is the average value of $Q_i$ at each time step.

## 4 Results

### 4.1 Simulation results of the coupled atmospheric-hydrologic systems

The coupled atmospheric-hydrologic systems mentioned in section3.3 were used to simulate the four storm events. The flood processes at different grid divisions were retrieved by the gridded atmospheric-hydrologic coupled system, and the comparison was also made with the results from the lumped atmospheric-hydrologic coupled system. As shown in Fig.11 and Table 5, the lumped system performed worse than the gridded system in most cases. The simulation results of the lumped system were close to the gridded system with different grid divisions for storm event 1 and 2, while the simulation results had a significant difference between lumped system and grid system for storm event 3 and 4. The gridded system, especially with the 1×1 km and 3×3 km grid, can improve accuracy and reduce error of the flood simulation for event 4.

The simulated results based on different grid sizes were similar for storm event 1, though the 9×9 km grid had the optimal simulation performance. All the coupled atmospheric-hydrologic systems at different grid divisions performed well for storm event 2, and 3×3 km grid produced the optimal simulation result. For storm event 3 (a short-duration heavy rainfall), although the peak flow error and flood volume error were not significant, the simulation results failed to truly reflect the runoff process. The 1×1 km grid generated the optimal simulation result for storm event 4.

[Figure 11 and Table 5]

It can be easily found that three grid sizes led to different simulation results for different rainfall events. The absolute value of $R_f$ was within 3.69% ~8.60% for 1×1 km grid, 4.69%~10.15% for 3×3 km grid and 6.09%~9.08% for 9×9 km grid. The absolute value of $R_l$ ranged from 4.40% to 11.81% for 1×1 km grid, ranged from 9.12% to 10.36% for 3×3 km grid, and ranged from 8.62% to 13.43%. The NSE is from 0.3669 to 0.8986 for 1×1 km grid, from 0.3232 to 0.8573 for 3×3 km grid, and from 0.3105 to 0.8302 for 9×9 km grid. Considering the rainfall simulation errors from WRF model for storm event 3, the 1×1 km and 3×3 km grids provided better simulation results overall while the 9×9 km grid led to unstable variations over a wide.



However, for a specific storm event, higher resolution may not lead to better flood process simulation. For example, the 9×9 km grid provided the optimal simulation results for storm event 1, while the best grid size was 3×3 km for rainfall event 2.

**4.2 Relationship between the rainfall evenness and the performance of the gridded model with different grid size**

Considering the spatial distribution characteristics of the rainfall, the gridded rainfall simulation results from WRF model were more suitable for the application of gridded Hebei model than lumped Hebei model. In order to analyse the relationship between the rainfall evenness and the performance of the gridded model with different grid size, the $C_v$ values were calculated for the rainfall simulations from WRF model. Table 6 showed that there were not much difference among the $C_v$ values with different grid size for the same storm event, while the difference was obvious for different storm events. The ranking of the spatial evenness of the rainfall simulations was event 1 > event 2 > event 3 > event 4.

**[Table 6]**

The average flood simulation results ($\overline{R_l}$, $\overline{R_f}$ and $\overline{NSE}$) for different grid size were calculated and used to compare with the flood simulation results ($R_{l\text{-lumped}}$, $R_{f\text{-lumped}}$ and $NSE_{\text{-lumped}}$) of the lumped system (shown in Table 7). Due to the large error of the rainfall simulation, storm event 3 was used as a reference event for further comparisons in this study. For other three storm events, the difference between the gridded system and lumped system became more significant as the heterogeneity of the rainfall spatial distribution increased. In other words, when the spatial distribution of the rainfall was uniform, as event 1 and 2, both the lumped and gridded systems achieved good simulation results. The simulation results of the former were even superior to those of the latter in some cases, such as the *NSE* for event 2. However, when the spatial distribution of the rainfall was uneven, the lumped system in most cases could only simulate the overall runoff situation but failed to fully describe the runoff process, while the gridded system could obtain better simulation such as for event 4. Furthermore, the 3×3 km grid produced the most stable simulation results for the event 1 and 2, which had the evenly distributed rainfall in space. It meant that the simulation results might not always be improved with higher dimension precision and more complicated system. As the spatial distribution of rainfall became uneven, the simulated effect with 9×9 km grid declined quickly and other grid sizes led to better flood process simulation. The simulation results tended to improve with higher dimension precision.

**[Table 7]**

**5 Discussion**

The errors of the rainfall simulation from WRF model have significant influence on the coupled atmospheric-hydrologic modeling system. Did the above conclusions change if the rainfall errors were eliminated? It was necessary to analyse the flood simulation results driven by the simulated rainfall, which was corrected by the measured values from the rain gauges. We assumed that the WRF model had the ability to reflect the spatial distribution of the rainfall. The Thiessen polygon method was used to divide the control region of each rain gauge. The areal rainfall of the control region based on the rain gauge was



regarded as the true rainfall value, which was allocated by the spatial distribution ratio from the simulated rainfall. The average value of the corrected gridded rainfall should be equal to the measured value of the rain gauge in a specific control region.

The corrected gridded rainfall was used to drive the coupled atmospheric-hydrologic modeling system and the simulation results were shown in Table 8. The system obtained similar simulation results with three different grid sizes for event 1 and 2.

It was hard to say which grid division was the most outstanding, and higher resolution may not lead to better flood process simulation. The corrected rainfall led to much better flood process simulations than the uncorrected rainfall from the WRF model. The simulations had significant improvement by using the corrected rainfall for event 3, and the system can reproduce the flood process. The *NSE*s were all above 0.88 and the $R_f$s were all lower than 3% at different grid sizes. The system with low $R_f$, $R_l$ and high *NSE*, also perform well with different grid divisions for event 4. Higher resolution can lead to better flood

process simulation for event 3 and 4. The system with 1×1 km grid size had the best simulation result. The conclusions were similar for both the corrected rainfall and the uncorrected rainfall. It was not always better for higher resolution, and the grid size selection had a great relationship with the rainfall evenness.

**[Table 8]**

## 6 Conclusion

This study established a couple atmospheric-hydrologic modeling system with variable grid sizes in sub-humid and sub-arid area in Northern China. The choice of coupling scales (1×1km, 3×3km, 9×9km) was disccused in depth. The WRF model and the gridded Hebei model are used to establish the gridded atmospheric-hydrologic coupled system. The lumped Hebei model was also used to establish the lumped atmospheric-hydrologic coupled system, and the simulation result served as a reference. Contrastive analyses of the flood process simulations from the gridded atmospheric-hydrologic coupled system and the lumped

atmospheric-hydrologic coupled system were carried out. Four main conclusions can be drawn: 1) the lumped system performed worse than the gridded system in most cases, while the simulation results were close to the gridded system for the storm events with uniform spatial distribution; 2) the coupled atmospheric-hydrologic systems at different grid divisions obtained similar simulation results and performed well for the storm events with uniform spatial distribution; 3) the simulation results might not always be improved with higher dimension precision, and the grid size selection should considering the

rainfall evenness; 4) for the storm events with uneven spatiotemporal distribution, the corrected rainfall can improve the simulation results significantly, and higher resolution can lead to better flood process simulation. The flood forecasting was the stress and difficulty in sub-humid and sub-arid area in Northern China. The couple atmospheric-hydrologic modeling system was influenced by the rainfall forecast accuracy and physics mechanism of hydrologic model. Further research considering the improvement of rainfall forecast and hydrologic model, should be carried out to further verify the conclusions

of this study.



## Author Contributions

All the authors contributed to the conception and the development of this manuscript. Jiyang Tian and Jia Liu contributed to establish the coupled atmospheric-hydrologic modeling system. Yang Wang and Wei Wang assisted in the calculations and the analyses. Chuanzhe Li and Chunqi Hu helped with the figure production and the manuscript writing.

## 5  Acknowledgements

This study was supported by the National Natural Science Foundation of China (51822906), the National Key Research and Development Project (2017YFC1502405), the Major Science and Technology Program for Water Pollution Control and Treatment (2018ZX07110001), and the IWHR Research & Development Support Program (WR0145B732017).

**Competing interests:** The authors declare that they have no conflict of interest.

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



**Table captions**

Table 1: Four 24h storm events selected from the study site.

Table 2: $C_v$ values for the four storm events.

Table 3: Basic settings of the WRF model for the two sub-watersheds.

Table 4: Physical parameterizations adopted for the selected storm events.

Table 5: Simulation results of the coupled atmospheric-hydrologic systems for four storm events.

Table 6: $C_v$ values of simulated rainfall spatial distribution for the four storm events.

Table 7: Comparation between gridded system and lumped system for flood simulation results.

Table 8: Simulation results of the coupled atmospheric-hydrologic systems based on the corrected gridded rainfall for four storm events.

**Figure captions**

Figure 1: Relationship between the model complexity and the goodness-of-fit of a data-driven model.

Figure 2: Rainfall-runoff processes of the four 24h storm events.

Figure 3: Location map of the two study sub-watersheds.

Figure 4: Topographic map of the two study sub-watersheds with the locations of rain and flow gauge stations.

Figure 5: DEM grids for the Fuping sub-watershed with channel and non-channel cells: (a) 1×1 km grids; (b) 3×3 km grids;

(c) 9×9 km grids..

Figure 6: DEM grids for the Zijingguan sub-watershed with channel and non-channel cells: (a) 1×1 km grids; (b) 3×3 km grids; (c) 9×9 km grids.

Figure 7: Main structure of the Hebei rainfall-runoff model built on each grid cell.

Figure 8: The water exchange processes in a non-channel grid cell.

Figure 9: The water exchange processes in a channel grid cell.

Figure 10: The coupled atmospheric-hydrologic system.

Figure 11: Flood process simulations of the coupled atmospheric-hydrologic systems for four storm events.





**Table 1.** Four 24h storm events selected from the study site.

| Event | Sub-watershed | Start time | End time | 24h accumulative rainfall (mm) | Peak discharges (m$^3$/s) |
|---|---|---|---|---|---|
| 1 | Fuping | 07/29/2007 20:00 | 07/30/2007 20:00 | 63.38 | 29.7 |
| 2 | Fuping | 07/30/2012 10:00 | 07/31/2012 10:00 | 50.48 | 70.7 |
| 3 | Fuping | 08/11/2013 07:00 | 08/12/2013 07:00 | 30.82 | 46.6 |
| 4 | Zijingguan | 07/21/2012 04:00 | 07/22/2012 04:00 | 155.43 | 2580.0 |





**Table 2.** $C_v$ values for the four storm events.

| Rainfall event | 1 | 2 | 3 | 4 |
|---|---|---|---|---|
| Spatial distribution | 0.3975 | 0.1927 | 0.7400 | 0.6098 |
| Temporal distribution | 0.6011 | 1.0823 | 2.3925 | 1.8865 |





**Table 3.** Basic settings of the WRF model for the two sub-watersheds.

| Parameter | Setting scheme |
|---|---|
| Power frame | Non-hydrostatic |
| Driving data | FNL |
| Driving data interval | 6 h |
| Spin-up time | 6 h |
| Integration time step | 6 s |
| Output data interval | 1 h |
| Grid center of Fuping sub-watershed | 39°04′15″N,113°59′26″E |
| Grid center of Zijingguan sub-watershed | 39°25′59″N,114°46′01″E |
| Horizontal grid system | Arakawa-C |
| Nesting scheme | Three layers of nested grids |
| Grid division of Fuping sub-watershed | Domain 1: 26×28<br>Domain 2: 42×48<br>Domain 3: 84×96 |
| Grid division of Zijingguan sub-watershed | Domain 1: 26×28<br>Domain 2: 42×48<br>Domain 3: 84×96 |
| Nesting ratio among the three layers of grids | 1:3 |
| Horizontal resolution | Domain 1: 9 km<br>Domain 2: 3 km<br>Domain 3: 1 km |
| Vertical stratification | 40 |
| Top-layer pressure | 50 hPa |
| Projection mode | Lambert |





**Table 4.** Physical parameterizations adopted for the selected storm events.

| Physical parameterisations | Storm event 1 | Storm event 2 | Strom event 3 | Storm event 4 |
|---|---|---|---|---|
| Microphysics | Lin | WSM6 | Lin | Lin |
| Cumulus parameterisation | KF | GD | KF | GD |
| Planetary boundary layer | MYJ | YSU | YSU | MYJ |
| Long/short wave radiation | RRTM/Dudhia | RRTM/Dudhia | RRTM/Dudhia | RRTM/Dudhia |
| Land surface model | Noah | Noah | Noah | Noah |





**Table 5.** Simulation results of the coupled atmospheric-hydrologic systems for four storm events.

| Storm event | Grid size | $R_l$ (%) | $R_f$ (%) | NSE |
|---|---|---|---|---|
| Event 1 | Lumped | 13.04 | 12.23 | 0.6012 |
| | 1×1 km | 11.81 | 8.60 | 0.8196 |
| | 3×3 km | 10.36 | 10.15 | 0.6111 |
| | 9×9 km | 8.62 | 7.60 | 0.8302 |
| Event 2 | Lumped | -15.23 | -8.98 | 0.8368 |
| | 1×1 km | -9.94 | -6.64 | 0.8302 |
| | 3×3 km | -9.12 | -6.48 | 0.8353 |
| | 9×9 km | -12.39 | -8.11 | 0.7728 |
| Event 3 | Lumped | -24.18 | -8.09 | 0.1524 |
| | 1×1 km | -4.40 | -3.69 | 0.3669 |
| | 3×3 km | -10.73 | -4.69 | 0.3232 |
| | 9×9 km | -9.59 | -6.09 | 0.3105 |
| Event 4 | Lumped | -20.87 | -14.69 | 0.6322 |
| | 1×1 km | -5.88 | -6.95 | 0.8986 |
| | 3×3 km | -9.60 | -8.04 | 0.8573 |
| | 9×9 km | -13.43 | -9.08 | 0.7107 |





**Table 6.** $C_v$ values of simulated rainfall spatial distribution for the four storm events.

| Grid size | Storm event | | | |
|---|---|---|---|---|
| | 1 | 2 | 3 | 4 |
| 1×1 km | 0.2105 | 0.1998 | 0.3932 | 0.3952 |
| 3×3 km | 0.2193 | 0.2081 | 0.4096 | 0.4117 |
| 9×9 km | 0.2700 | 0.2258 | 0.4423 | 0.4570 |



**Table 7.** Comparation between gridded system and lumped system for flood simulation results

| Storm event | $\bar{R}_l$ | $\bar{R}_l - R_{l\text{-lumped}}$ | $\bar{R}_f$ | $\bar{R}_f - R_{f\text{-lumped}}$ | $\overline{NSE}$ | $\overline{NSE} - NSE_{\text{-lumped}}$ |
|---|---|---|---|---|---|---|
| Event 1 | 10.26 | 2.78 | 8.78 | 3.45 | 0.7536 | 0.1524 |
| Event 2 | -10.48 | 4.75 | -7.08 | 1.90 | 0.8128 | 0.0240 |
| Event 3 | -8.24 | 15.94 | -4.82 | 3.27 | 0.3335 | 0.1811 |
| Event 4 | -9.64 | 11.23 | -8.02 | 3.96 | 0.8222 | 0.1900 |



**Table 8.** Simulation results of the coupled atmospheric-hydrologic systems based on the corrected gridded rainfall for four storm events.

| Storm event | Grid size | $R_l$ (%) | $R_f$ (%) | NSE |
|---|---|---|---|---|
| | 1×1 km | 2.42 | -1.54 | 0.9027 |
| Event 1 | 3×3 km | 3.34 | -2.31 | 0.8901 |
| | 9×9 km | 3.26 | -1.49 | 0.9087 |
| | 1×1 km | -3.37 | -0.87 | 0.9266 |
| Event 2 | 3×3 km | -3.16 | -0.08 | 0.9237 |
| | 9×9 km | -2.5 | 1.25 | 0.9227 |
| | 1×1 km | -2.12 | -2.61 | 0.9026 |
| Event 3 | 3×3 km | -2.77 | -2.62 | 0.8891 |
| | 9×9 km | -8.4 | -2.53 | 0.8843 |
| | 1×1 km | -0.26 | -1.06 | 0.9287 |
| Event 4 | 3×3 km | -2.47 | -1.16 | 0.9233 |
| | 9×9 km | -4.23 | -2.12 | 0.9118 |



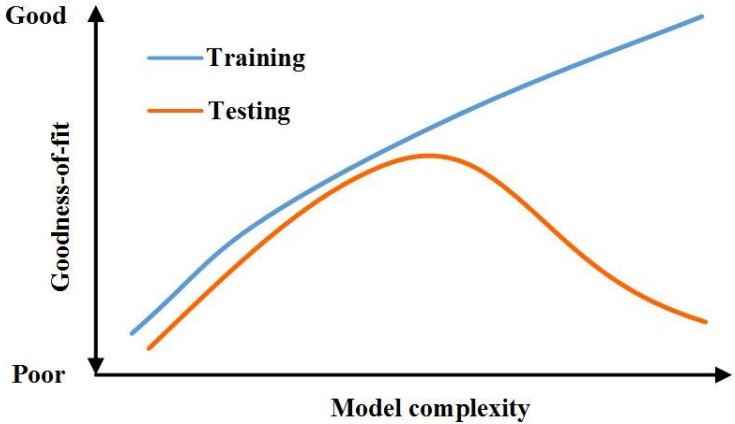

**Figure 1.** Relationship between the model complexity and the goodness-of-fit of a data-driven model.





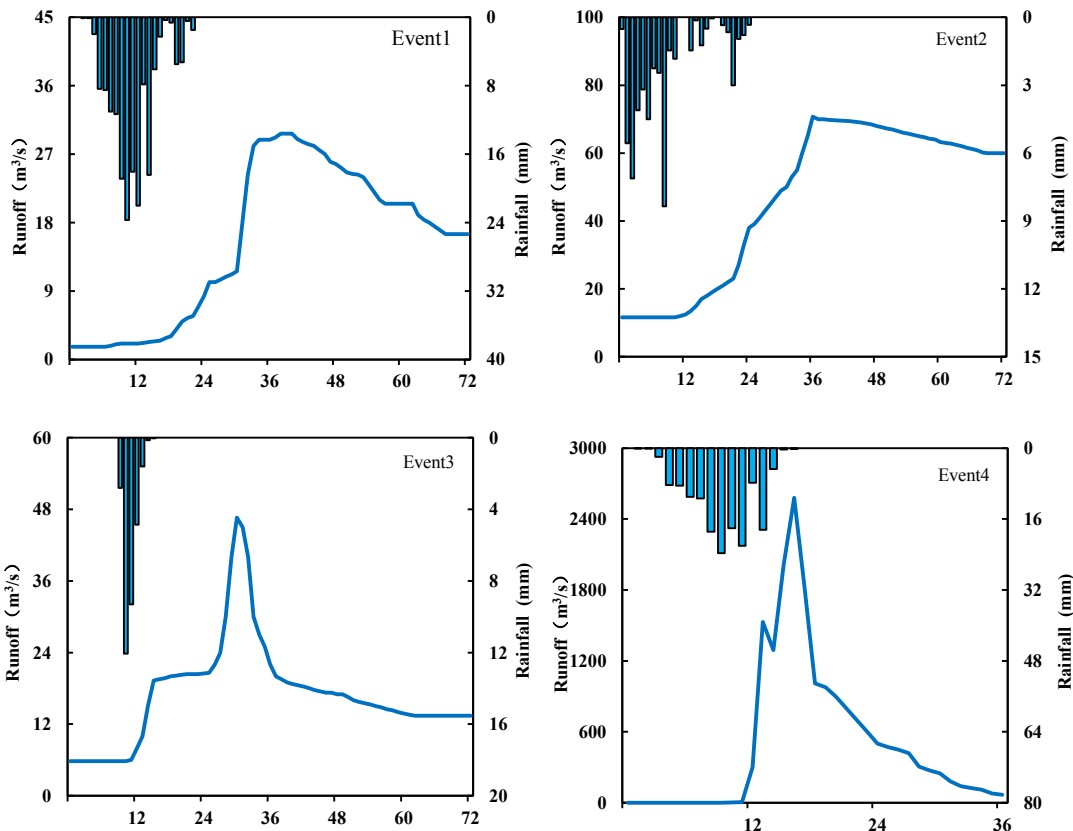

**Figure 2.** Rainfall-runoff processes of the four 24h storm events.



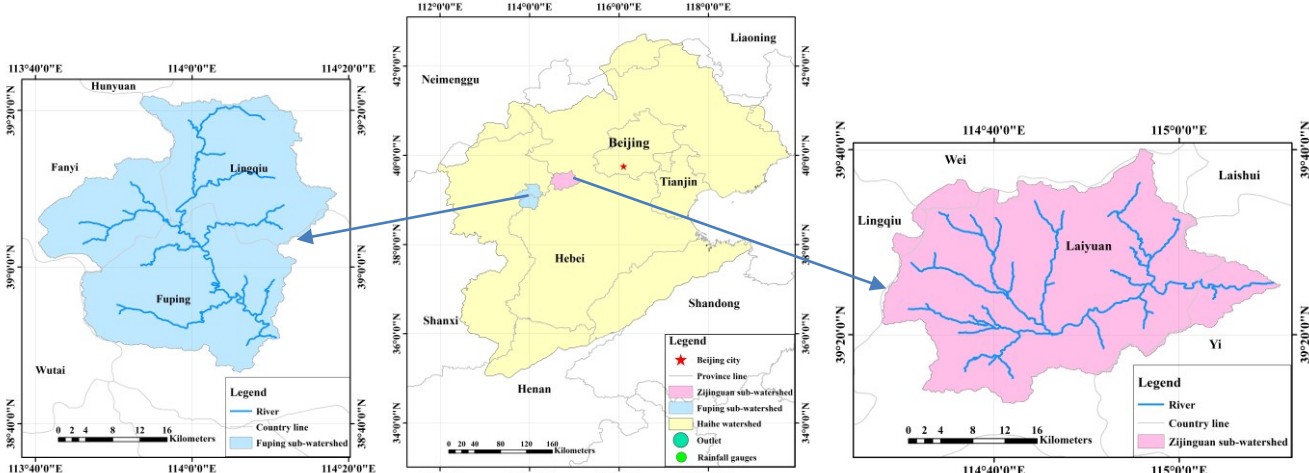

**Figure 3.** Location map of the two study sub-watersheds.


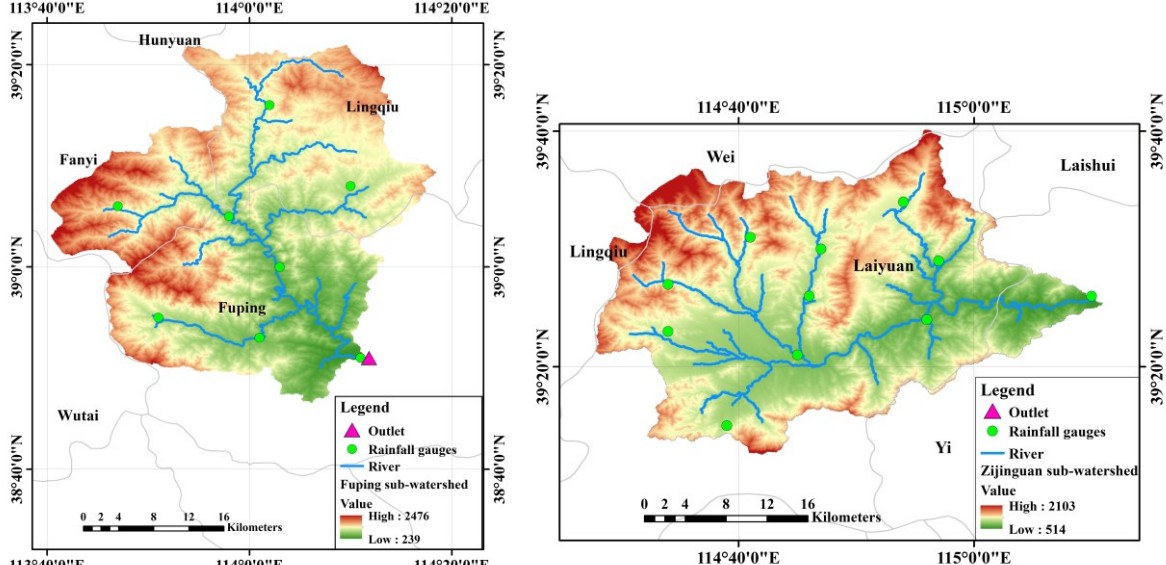

**Figure 4.** Topographic map of the two study sub-watersheds with the locations of rain and flow gauge stations.



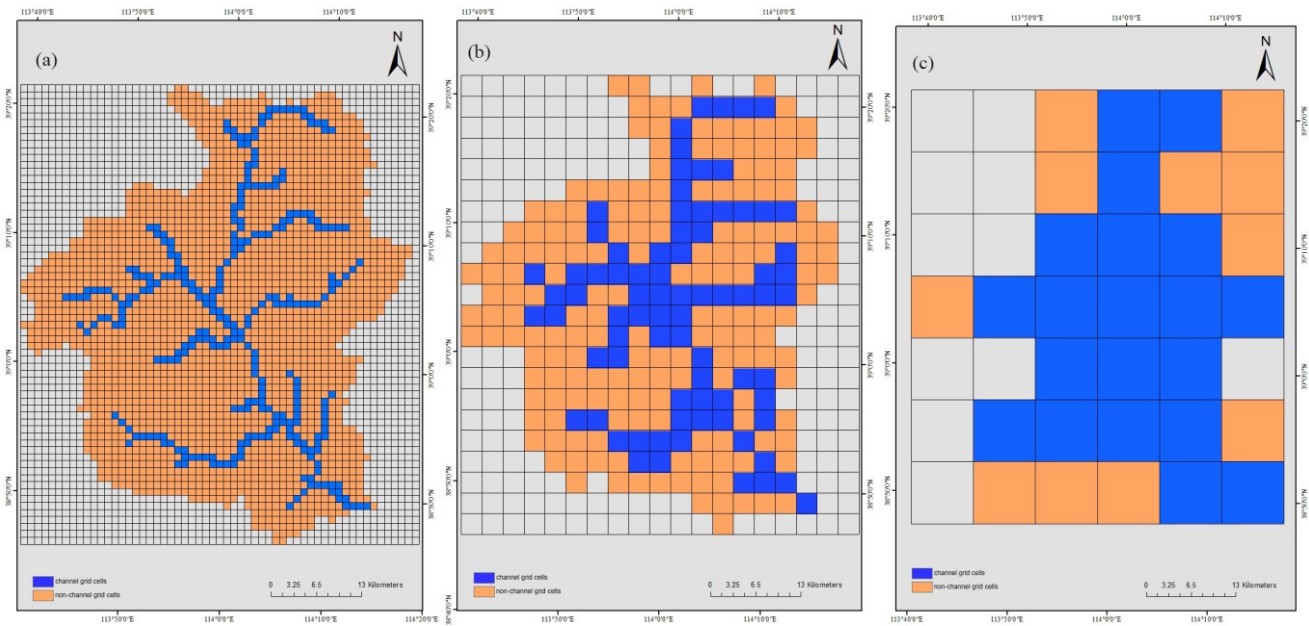

**Figure 5.** DEM grids for the Fuping sub-watershed with channel and non-channel cells: (a) 1×1 km grids; (b) 3×3 km grids; (c) 9×9 km grids.



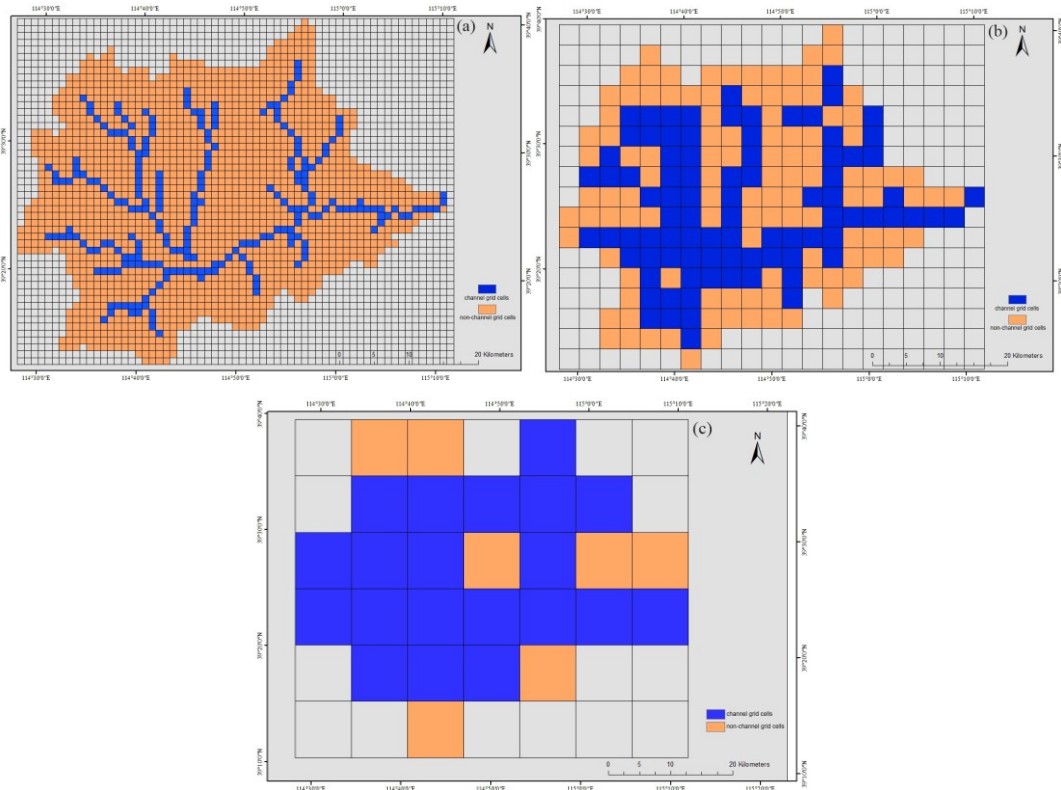

**Figure 6.** DEM grids for the Zijingguan sub-watershed with channel and non-channel cells: (a) 1×1 km grids; (b) 3×3 km grids; (c) 9×9 km grids.


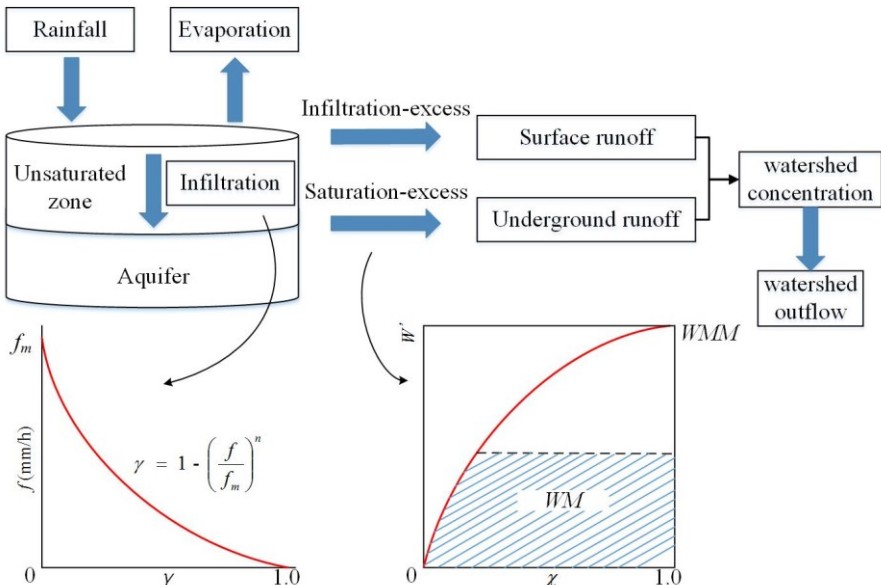

**Figure 7.** Main structure of the Hebei rainfall-runoff model built on each grid cell.



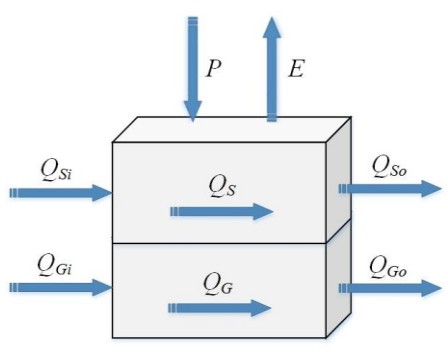

**Figure 8.** The water exchange processes in a non-channel grid cell.



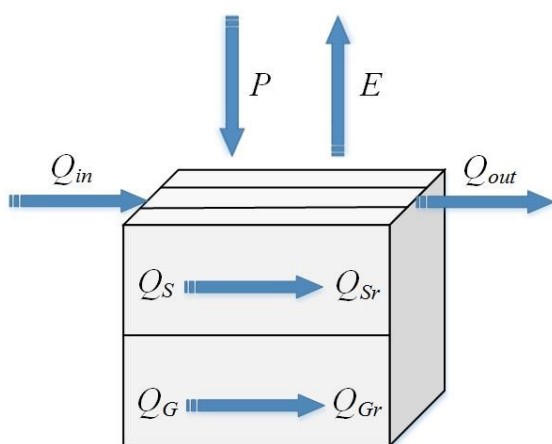

**Figure 9.** The water exchange processes in a channel grid cell.





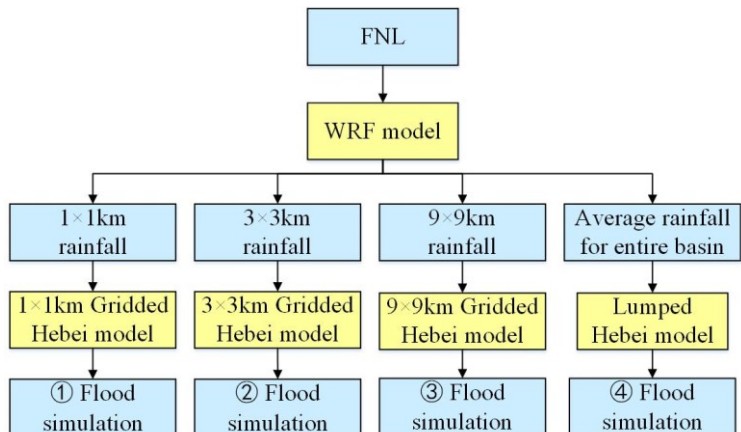

**Figure 10.** The coupled atmospheric-hydrologic system.





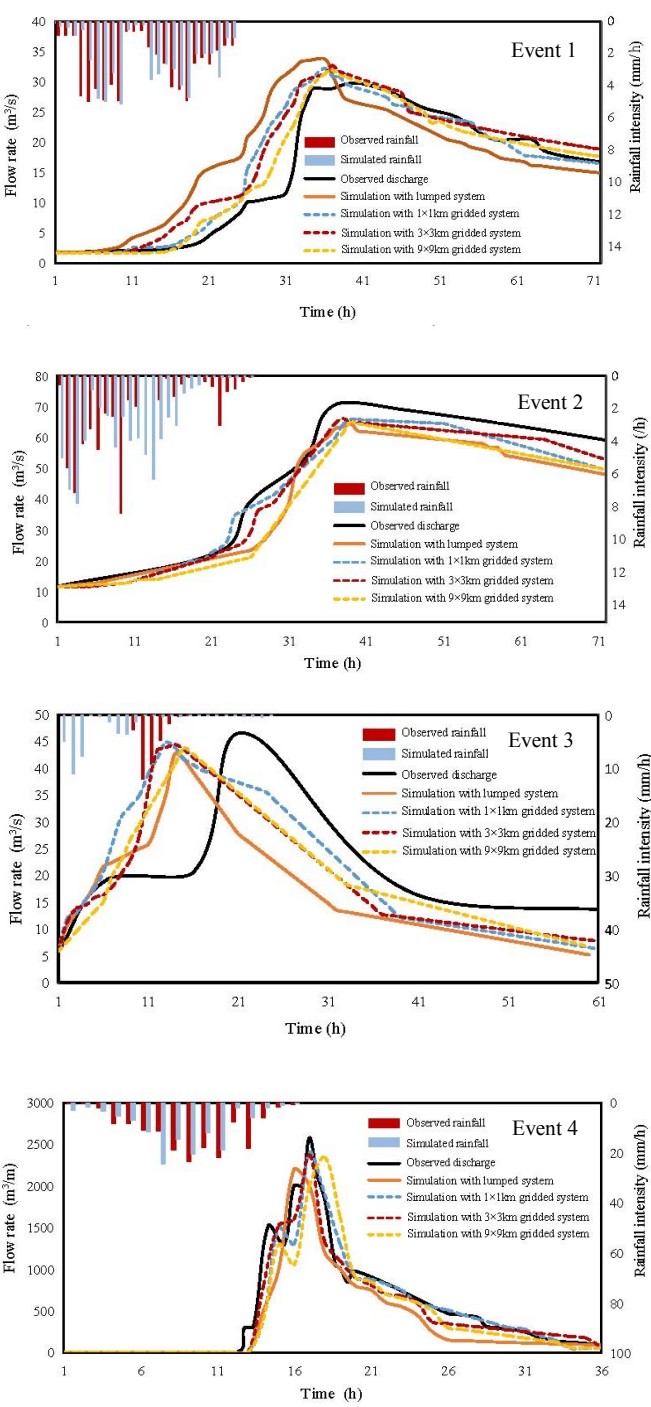

**Figure 11.** Flood process simulations of the coupled atmospheric-hydrologic systems for four storm events.