# Peer review of "A coupled atmospheric-hydrologic modeling system with variable grid sizes for rainfall-runoff simulation in semi-humid and semi-arid watersheds: How does the coupling scale affects the results?"

_Hydrology and Earth System Sciences, 2019_

## Referee Comment (RC1) · Anonymous Referee #1 · 9 Mar 2020

The paper presents a study to explore the impact of coupling an atmospheric model with a hydrological model (lumped and distributed) using different model grid sizes for the simulation of river flows. The results showed that for precipitation events with uneven spatiotemporal distribution, a higher resolution model can lead to better flow simulations, whereas for events with uniform spatial distribution, the coupling scale has less impact on flow simulation and in this case a lumped model performs as good as a distributed model. The paper is interesting and well written with some minor spelling

mistakes, but there are a number of questions that the authors need to address before the paper is recommended for publication.

1- The paper is essentially divided in two parts: a) A coupled atmospheric-hydrologic system with variable grid sizes for rainfall-runoff simulation and b) the development of a new distributed hydrological model. The distributed model helps to answer the research question posed in the first part (How coupling affect the results?). However, the authors are actually presenting a new distributed hydrological model (why?) rather than using a distributed model widely accepted by the hydrological community. why do we need a new model? In order to answer your research question, you need to demonstrate that your hydrological model works well for a number of storms and that the model has been properly calibrated. I think this area is a bit weak in the paper and needs further results. For instance, the authors calibrate the model with 7 storms, but they do not give any indication on how the calibration was achieved, what type of storms are used and so on. Usually to calibrate hydrological models, continuous rainfall-runoff time series are needed rather than individual events in order to account for initial conditions in the model such as soil moisture, catchment wetness, etc. In addition, the authors quoted a model calibration efficiency (NSE) of 0.686, but three out of four events used in the validation showed an efficiency higher than 0.75. Normally the performance of the model in the validation phase is worse than in the calibration, but this is not the case in your analysis. why? For the storms used for validation, how do you account for the initial catchment conditions? It is obvious that if the model starts completely dry, the results will be affected by this even if the model is calibrated properly. please expand on this. In addition, it is unclear from all the equations used to describe the model, which are model parameters. maybe you can summarise all the model parameters in a table and include their range of values. what about model parameter calibration uncertainty? What ranges of model parameters did you use and why? It is well known that different parameter sets can produce a similar model performance (equifinality). You need to look at parameter uncertainty and maybe produce an ensemble of hydrographs rather than a deterministic one. what about the uncertainty in the observations (e.g. rain

gauges, flow stations, etc)? You did not mention any of this in the paper. 2- The second point is about the use of different WRF microphysics parameterisations (MP). Table 4 shows that different MPs are used to simulate each storm event. How do you isolate the impact of the WRF MPs in your results? How do you justify that the difference in the results is due to the different grid resolutions rather than the fact that different MPs are used to simulate each event? What is the performance of WRF simulating those storms?

Other comments

The abstract talks about the "Hebei model", but this model is not known in the literature and you have not introduced this model yet. In page 3: "a higher error rate". Do you mean "larger errors"? the use of "error rate" might be confusing. "variation pattern" - again unclear what you mean here. "Hebei model" Is this model published? If so, you need to include a proper reference. If not, then you should describe the model in the methodology and do not use "Hebei model" until this has been described. "1x1km ... 9x9km" Are these spatial scales within the WRF model domains or are you talking about the spatial scales of the hydrological model? Section 2. The description of the catchments should be placed before the description of the events otherwise how do you know which catchment outlet are you talking about here? Is Cv a standard metric to characterise the spatial and temporal distribution of precipitation processes? If so, you need to include a reference. I believe to characterise the spatial distribution of precipitation is better to use semi variograms, correlograms or by looking at the spatial correlation of the precipitation field. Likewise, with the temporal correlation. Could you please expand and justify why a simple metric like this was used? Cv here is highly dependent on the number of rain gauge stations available. Is the WRF rainfall field used to compute Cv? table 2 - I think you need to explain what values of Cv correspond to a highly variable event in space and time. how did you come up with the critical values of Cv (0.4 and 1 for spatial/temporal distribution respectively)? you need to justify these values. Section 2.2 can you provide more information of these catchments (e.g. catchment descriptors)? e.g. apart from catchment area, mean annual rainfall, catchment slope, mean flow, predominant land use and soil types, type of geology, percentage of urban area, etc. you can summarise all this info in a table. this is important to understand the catchment response to precipitation. Section 3.1.1. "in consistence" consistent? The grid resolution of the coarser domain is 9km, but the NCEP analysis is about 100km. How does the WRF model handles this discrepancy with the initial and boundary conditions between the outer domain and the analysis? What's the temporal resolution of the analysis? Section 3.1.2. Need to define all acronyms for the physical parameterisations. Section 3.2.2 ". . . has widely been applied in Hebei Province . . ." add references. Summarise all model parameters in a table with the range of values for calibration. How do you account for different soil types, land use, etc? These will have important implications in terms of runoff production. Do you use ET (evapotranspiration) or only E (evaporation)? do you have forest in any of the basins? Expand on the calibration and validation of the hydrological model, including both, lumped and distributed models. fig 11. Unclear if these results are for catchment A or B. Table 5. how do you isolate the impact of the different WRF microphysics parameterisations in your results?

---

## Referee Comment (RC2) · Anonymous Referee #2 · 27 Mar 2020

This study examined the optimum scale of an atmospheric-hydrologic coupled system for rainfall-runoff simulation. The authors concluded that unnecessary complexity during model construction can result in an overfitting, which is quite well acknowledged by the hydrological community for the data-driven models, but not yet for the physically-based ones. Generally, I think that this study is well-designed, and the analyses presented in the manuscript are thorough to address the objective of the study. However, I have some concerns and questions related to the performance of the coupled system and the possible uncertainty associated with the experiments. The authors should clarify the issues provided below before the manuscript is considered for publication in HESS.

Major concerns:

1. More information of the Hebei model and the calibration of its parameters should be given. What is the advantage of using the Hebei model in the study area? What are the parameters calibrated in section 3.2 and how are they calibrated? Why choosing the 7 floods in Fuping and 6 floods in Zijingguan to calibrate the model? What are the values of the calibrated parameters finally used in the coupled system?

2. I think the gridded Hebei model is a semi-distributed model. The main goal of establishing the gridded Hebei model is to match the rainfall simulation from the NWP system. Hence, the Hebei model does not consider the spatial variability of the underlying condition of the watersheds. If so, I do not quite understand why the soil storage capacity and the infiltration capacity is discretized across the grid cells?

3. The errors of the coupled system generally come from two parts: the NWP system and the hydrologic model. Since the WRF model is used (the rainfall error of which is normally quite considerable), I believe the accuracy of the simulated rainfall is the main factor affecting the performance of the coupled system (although there might also be uncertainties from the hydrologic model). Could the authors specify the rainfall errors from each storm events and quantity how much the system errors come from the rainfall simulations? A further question is, how to improve the simulated rainfall from the NWP system in order to improve the performance of the coupled system. For example, some grid-based observations, such as QPEs from the weather radar might be helpful.

4. I agree that Cv is used to describe the evenness of rainfall for both spatial and temporal distributions. However, a critical value of 0.40 for evenness in space and 1.00 for evenness in time, is hard to follow. Explain how the threshold is obtained. You can say event 1 has relatively even distributed rainfall according to Cv rather than using the

value of 0.4 as a threshold.

5. It is concluded from the study that for storm events with uneven rainfall distributions, a finer coupling scale can lead to a better performance of the coupled system, however, the coupling scale shows less impact on the system for events with uneven distributions. To my opinion, these conclusions are highly dependent on the case studies. Considering the study only focusing on two semi-humid and semi-dry watersheds with limited storm events involved, it is better to point out that the results are some kind of site-specific. More case studies are needed before more general conclusions can be achieved.

Spelling and grammar mistakes should be checked carefully throughout the manuscript.

Page 2, line 5 and line 9: "atmosphere-hydrologic" should be "atmospheric-hydrologic" Page 5, line 11: "Based on the historical storm events in the study area, and using 5% as a cutoff" should be "Based on the historical storm events in the study area by using 5% as a cutoff" Page 10, line 21: "Grid center coordinates. . .for driving the hydrologic model" should be "The coordinates of the grid cell centers . . .to drive the hydrologic model" Page 11, line 22: ". . .three grid sizes led to different simulation results for different rainfall events" should be ". . .different rainfall events have different simulation results with the three grid sizes". Page 12, line 4: "Considering the spatial distribution characteristics of the rainfall. . ." should be "Considering the characteristics of the spatial rainfall distributions. . ." Page 12, line 29: ". . .the WRF model had the ability to reflect the spatial distribution of the rainfall" should be ". . .the WRF model was able to capture the spatial patterns of the simulated rainfall. . ." Page 13, line 4 ". . .similar simulation results with three different grid sizes. . ." should be ". . .similar simulation results of the three different grid sizes. . ."

---

## Author Comment (AC1) · 10 Apr 2020

We appreciate very much the referee's insightful comments and helpful suggestions for our manuscript. Efforts have been made to address every point of the referee's concerns. During the revision being carried on, we are also encouraged by the positive comments from the referee ". . .the paper is interesting and well written with some minor spelling mistakes. . ." Gramma mistakes and spelling errors will carefully be checked

before the revision is finally submitted. With the help of the referee, we hope the revised manuscript can be found rigorously and sufficiently improved.

Main comments: Point 1: The authors are actually presenting a new distributed hydrological model (why?) rather than using a distributed model widely accepted by the hydrological community. why do we need a new model? Reply: Thanks for the reviewer's question. The Hebei rainfall-runoff model is specially developed to describe the runoff generation mechanisms in the semi-humid and semi-dry area of Northern China, which has been successfully applied in Hebei Province for rainfall-runoff modeling and real-time flood forecasting. Due to the perennial water shortage and groundwater over-exploitation, both storage-excess and infiltration-excess is found with great seepage along the river channel during the storm season. The obvious advantage of the Hebei model is the consideration of both storage-excess and infiltration-excess mechanisms for rainfall-runoff generation. It is a well-known conceptual model in China, as popular as the Xin'anjinag model. The model is easily used, and can widely be applied to other semi-humid and semi-arid watersheds with complicated (both storage-excess and infiltration-excess) mechanisms for rainfall-runoff generation. The description for the storage-excess part in the Hebei model is the same as the Xin'anjiang model. On the other hand, to reflect the heterogeneity of the infiltration capacity across the catchment, a distribution curve is adopted and expressed as Eqn. (2). The Horton infiltration model is also applied to obtain the infiltration volume for the river channel seepage. When the calculation interval is one hour, the infiltration volume can be calculated by the Eqn. (16). In order to clarify this issue, the following sentences are added in Line 22-28, Page 6: "Due to the perennial water shortage and groundwater overexploitation, both storage-excess and infiltration-excess is found in the study area with great seepage along the river channel during the storm season. The obvious advantage of the Hebei model is the consideration of both storage-excess and infiltration-excess mechanisms for rainfall-runoff generation. The model is easily applied and can be used in other semi-humid and semi-arid watersheds. In the Hebei model, the description for the storage-excess part is the same as that in the Xin'anjiang model. On the other

hand, the infiltration capacity across the watershed is described by a distribution curve described below, and the Horton model is applied to calculate the seepage along the river channel during the river routing."

Point 2: In order to answer your research question, you need to demonstrate that your hydrological model works well for a number of storms and that the model has been properly calibrated. I think this area is a bit weak in the paper and needs further results. For instance, the authors calibrate the model with 7 storms, but they do not give any indication on how the calibration was achieved, what type of storms are used and so on. Usually to calibrate hydrological models, continuous rainfall-runoff time series are needed rather than individual events in order to account for initial conditions in the model such as soil moisture, catchment wetness, etc. In addition, the authors quoted a model calibration efficiency (NSE) of 0.686, but three out of four events used in the validation showed an efficiency higher than 0.75. Normally the performance of the model in the validation phase is worse than in the calibration, but this is not the case in your analysis. why? For the storms used for validation, how do you account for the initial catchment conditions? It is obvious that if the model starts completely dry, the results will be affected by this even if the model is calibrated properly. please expand on this.
Reply: We agree with the referee that the model calibration part needs to be strengthened in the manuscript. The SCE-UA (Shuffle Complex Evolution) method is used to calibrate the parameters of the Hebei model (Duan et al., 1994). Actually, we have very limited choices when selecting the calibration data. Considering the semi-humid and semi-dry conditions of the study area, the soil is relatively dry before the storm season, and there is not many storm events leading to significant peak discharges. In this case, 7 storm events in Fuping and 6 storm events in Zijingguan are chosen to calibrate the model, and 2 from each sub-watersheds are used to validate the model. Detailed information (i.e., the cumulative rainfall amounts and the peak discharges) of the events are summarized in the tables below. Considering there are already many table in the manuscript, these tables are not shown. When calibrating the model, the calibration events are bounded together to calculate one NSE value as the objective

function. In order to guarantee reasonable values for the initial model conditions, the 24-h storm event is not independently used, but with a continuous antecedent period of data with the length of 15-days before the start of the event. In this sense, the events used for calibrating the model is some kind of "continuous" time series data. As for the calibration and validation values, it should be clarify firstly that the NSE value of 0.686 we quoted in Section 3.3 is for model verification (validation), not for calibration. This is an average value resulted from the 2 validation events from each sub-watershed, indicating that the calibrated model is reliable for further applications in the study area. It is also noted by the referee that there are three out of the four events with a NSE value higher than 0.75 in Table 9. The four storm events used for testing the grid sizes are different from those used for calibration and validation, but as "further applications". Moreover, the validation results are from the lumped Hebei model, whereas the results of the four storm events are grid-based averaged from different grid sizes (each grid establishing a lumped model using the calibrate parameters). Therefore, the NSE values of the four events (some are higher than 0.75) are not comparable to those of the validation events (an average of 0.686). The following sentences are added in Line 29, Page 10 and Line 1-7, Page 11 to supplement more details about the model calibration and validation: "The SCE-UA (Shuffle Complex Evolution) method (Duan et al., 1994) is used to calibrate the parameters and the calibrated values are shown in Table 6. Due to the limited observational data, 7 storm events in Fuping and 6 storm events in Zijingguan are selected and used to calibrate the Hebei model, and another 2 from each sub-watersheds are used for model validation. In order to guarantee reasonable values for the initial model conditions, the storm events are not independently used, but with an antecedent period of data with the length of 15-days before the start of the event. The validation results show an average NSE value of up to 0.686, indicating the calibrated models are reliable for further applications. It should be noted that the four storm events in Section 2.2 are different from those used for calibration and validation."

Reference: Duan, Q., Sorooshian, S., Gupta, V. K. Optimal use of the SCE-UA global optimization method for calibrating watershed models, J. Hydrol., 158(3-4), 265-284,

doi: 10.1016/0022-1694(94)90057-4, 1994.

Point 3: In addition, it is unclear from all the equations used to describe the model, which are model parameters. maybe you can summarise all the model parameters in a table and include their range of values. what about model parameter calibration uncertainty? What ranges of model parameters did you use and why? It is well known that different parameter sets can produce a similar model performance (equifinality). You need to look at parameter uncertainty and maybe produce an ensemble of hydrographs rather than a deterministic one. what about the uncertainty in the observations (e.g. rain. gauges, flow stations, etc)? You did not mention any of this in the paper. Reply: Thanks for the reviewer's suggestion. In the revised manuscript, efforts are made to explain all parameters and the uncertainty. Firstly, a new table below (Table 6) is added to show the calibrated parameter values on Page 6. The ranges of the parameter values are determined based on the application experiences of the Hebei model, which has been used in Northern China for more than two decades. The importance of hydrological uncertainty analysis has been emphasized in recent years and there is a necessity to incorporate parameter uncertainty estimation wherever a hydrological model is used. However, the parameter uncertainty estimation needs considerable observational data (Hughes et al., 2010). As mentioned above, there are not enough historical storm-flood processes for us to use in the study area, which makes the estimation work rather difficult. In this study, the research focus is how the coupling scale affects the flow results from the atmospheric-hydrologic coupling system, and we believe the conclusions from the comparative analyses would be quite similar even if the ensemble simulations with perturbed model parameters were carried out. Nevertheless, the parameter uncertainty estimation and ensemble simulations are suggested in the future study when sufficient observational data are available. The following paragraphs are added to address this issue in Line 6-9, Page 14: "It should be mentioned that there is a necessity to incorporate parameter uncertainty analysis in this study. However, this will need a considerable set of the observational data (Hughes et al., 2010). Due to the lack of sufficient historical storm-flood processes, it is impossible

to carry out such analyses. Nevertheless, parameter uncertainty estimations and ensemble simulations with perturbed parameters are suggested in the future study when sufficient observational data are available." Reference: Hughes, D. A., Kapangaziwiri, E., Sawunyama, T. Hydrological model uncertainty assessment in southern Africa, J. Hydrol., 387, 221–232, doi: 10.1016/j.jhydrol.2010.04.010, 2010.

As for the uncertainty of the observations, the rainfall and flow data are estimated before the establishment of the coupled atmospheric-hydrologic system. Hourly rainfall data are obtained from the rain gauges in the two sub-watersheds. Although the density of the rain gauges is a little sparse, the accuracy of the observations has been verified by the radar data, which can be found in our previous study (Liu et al., 2018). The hydrologic station observes the flow data at discrete time intervals (at least once an hour), which becomes more intense during the flood season to guarantee more realistic flow responses. The following sentence is added in Line 8-10, Page 4: "Before the establishment of the coupled atmospheric-hydrologic system, quality controls of the observational rainfall-runoff data are carried out. Rainfall observations from the rain gauges are verified by the weather radar and interpolations are done to guarantee the continuity of the flow observations (Liu et al., 2018)." Reference: Liu, J., Tian, J., Yan, D., et al. Evaluation of Doppler radar and GTS data assimilation for NWP rainfall prediction of an extreme summer storm in northern China: from the hydrological perspective, Hydrol. Earth Syst. Sci., 22, 4329–4348, doi: 10.5194/hess-22-4329-2018, 2018.

Point 4: The second point is about the use of different WRF microphysics parameterisations (MP). Table 4 shows that different MPs are used to simulate each storm event. How do you isolate the impact of the WRF MPs in your results? How do you justify that the difference in the results is due to the different grid resolutions rather than the fact that different MPs are used to simulate each event? What is the performance of WRF simulating those storms? Reply: Actually in this study, the impacts of different coupling scales are compared for each certain storm event with the same MPs. The

comparison is not carried out among the four events with different MPs. In order to eliminate the modeling errors caused by choosing inappropriate WRF parameterisations, the most suitable physical parameterisations resulting the most realistic rainfall simulations are used for each of the four storm event (as shown in Table 5). The reason why these physical parameterisations are the best choices has been discussed in detail in our previous study (Tian et al., 2017a). The following sentence is added in Line 27-31 Page 5 to address this issue: "In order to eliminate the modeling errors caused by choosing inappropriate WRF parameterisations, the most suitable physical parameterisations resulting the best rainfall simulations (Tian et al., 2017a) are used for each of the four storm events, as shown in Table 5. It should be clarified that the comparison of different coupling scales is carried out for each of storm event under the same MPs, thus using different MPs for different events will not cause difficulties in analyzing the final results." Reference: Tian, J., Liu, J., Wang, J., et al. A spatio-temporal evaluation of the WRF physical parameterisations for numerical rainfall simulation in semi-humid and semi-arid catchments of Northern China, Atmos. Res., 191, 141–155, doi: 10.1016/j.atmosres.2017.03.012, 2017a.

Other comments: Point 5: The abstract talks about the "Hebei model", but this model is not known in the literature and you have not introduced this model yet. Reply: The introduction of the Hebei model is added in the abstract Line 14-16 Page 1: "The Hebei model is a conceptual rainfall-runoff model designed to describe a mixed runoff generation mechanism, including both storage-excess and infiltration-excess, in the semi-humid and semi-dry area of Northern China."

Point 6: In page 3: "a higher error rate". Do you mean "larger errors"? the use of "error rate" might be confusing. "variation pattern" - again unclear what you mean here. Reply: Revised accordingly. The sentences in page 3 are revised as below: "...but the accuracy of the rainfall information decreases such that the final simulation results may be subject to larger errors. Therefore, finding the underlying law of how the performance of the coupled atmospheric-hydrologic system is impacted by the coupling

scale is of great importance in enhancing the accuracy of rainfall-runoff simulation."

Point 7: "Hebei model" Is this model published? If so, you need to include a proper reference. If not, then you should describe the model in the methodology and do not use "Hebei model" until this has been described. Reply: The "Hebei model" is a published model. There have already been many literatures in Chinese, but quite rare in English. The authors recently have published some English work, which gives a detail description of the Hebei model. The following reference is added in Line 13, Page 3 and the sentence in Line 13-14 is also revised: "...the Hebei model (Tian et al., 2019), a conceptual model with mixed runoff generation mechanisms of both saturation-excess and infiltration-excess, is used to construct the gridded hydrologic model." Reference: Tian, J., Liu, J., Yan, D., et al. Ensemble flood forecasting based on a coupled atmospheric-hydrological modeling system with data assimilation, Atmospheric Research, 224, 127-137, doi: 10.1016/j.atmosres.2019.03.029, 2019.

Point 8: "1x1km...9x9km" Are these spatial scales within the WRF model domains or are you talking about the spatial scales of the hydrological model? Reply: "1×1 km, 3×3 km, and 9×9 km" are the coupling scales for the coupled atmospheric-hydrologic system. Therefore, they are not only the spatial scales of the WRF model outputs (three domains with the grid cell size being 1×1 km, 3×3 km, and 9×9 km), but also the spatial scales of the gridded Hebei model.

Point 9: Section 2. The description of the catchments should be placed before the description of the events otherwise how do you know which catchment outlet are you talking about here? Reply: Agreed and the order of section 2.1 and 2.2 is changed.

Point 10: Is Cv a standard metric to characterise the spatial and temporal distribution of precipitation processes? If so, you need to include a reference. I believe to characterise the spatial distribution of precipitation is better to use semi variograms, correlograms or by looking at the spatial correlation of the precipitation field. Likewise, with the temporal correlation. Could you please expand and justify why a simple metric

like this was used? Cv here is highly dependent on the number of rain gauge stations available. Is the WRF rainfall field used to compute Cv? Reply: Cv is a standard metric to describe the dispersion of measures, thus is used in this study to describe the evenness of rainfall distribution. The main advantage of the statistic is that the evenness of rainfall distribution can easily be quantified in both time and space by following the proposed rules in Section 2.2. We have a series of publications regarding the WRF model and the simulation of storm events with different spatio-temporal evenness, where the same Cv statistic and calculation rules are adopted (Tian et al., 2017a, b). We have also found other studies using the same statistic to describe the rainfall distributions (Yue et al., 2014; Jha et al., 2015). It is true that the statistics may depend on the number of rain gauges (actually any statistic may involve this uncertainty), but this is the best we can do. Considering the WRF simulations are not the "ground truth", and in this study observations from the rain gauges are used to evaluate the WRF simulations, hence the rainfall evenness are obtained from the rain gauges rather than the WRF simulations. The following references are added in Section 2.2 when introducing the use of Cv as an evaluation of the rainfall distribution evenness. References: Tian, J., Liu, J., Wang, J., et al. A spatio-temporal evaluation of the WRF physical parameterisations for numerical rainfall simulation in semi-humid and semi-arid catchments of Northern China, Atmos. Res., 191, 141–155, doi: 10.1016/j.atmosres.2017.03.012, 2017a. Tian, J., Liu, J., Yan, D., et al. Numerical rainfall simulation with different spatial and temporal evenness by using a WRF multiphysics ensemble, Nat. Hazards Earth Syst. Sci., 17, 563-579, doi: 10.5194/nhess-17-563-2017, 2017b. Yue, B. J., Shi, Z. H., Fang, N. F. Evaluation of rainfall erosivity and its temporal variation in the Yanhe River catchment of the Chinese Loess Plateau, Nat. Hazards, 74, 585-602, doi: 10.1007/s11069-014-1199-z, 2014. Jha, S. K., Zhao, H., Woldemeskel, F. M., et al. Network theory and spatial rainfall connections: An interpretation, J Hydrol., 527, 13-19, doi: 10.1016/j.jhydrol.2015.04.035, 2015.

Point 11: table 2 - I think you need to explain what values of Cv correspond to a highly variable event in space and time. how did you come up with the critical values of Cv

(0.4 and 1 for spatial/temporal distribution respectively)? you need to justify these values. Reply: In this study, the spatial and temporal Cv of the historical storms from 1985 to 2018 is calculated to analyse the characteristics of the rainfall evenness. A threshold of 5% is used to separate even and uneven storms. It is found that the storm events with a spatial Cv < 0.4 or with a temporal Cv < 1.0 account for 5% of the total storm events from 1985 to 2018. The methodology is also adopted in our previous publication (Tian et al., 2017a). However, the critical values of 0.4 and 1.0 are based on statistical analyses of historical storm events, thus are not transferable to other areas with different meteorological conditions. In order to avoid misunderstanding, the description part of the critical values are removed in the revised manuscript. Instead, the spatial and temporal evenness of rainfall distribution is ranked among different storm events. The following sentences can be found in Line 22-24 Page 4: "The smaller is the value of Cv, the more even is the rainfall distribution in space or time. According to Table 3, the ranking of the distribution evenness of rainfall in space is event 2 > event 1 > event 4 > event 3 and that in time is event 1 > event 2 > event 4 > event 3." Reference: Tian, J., Liu, J., Wang, J., et al. A spatio-temporal evaluation of the WRF physical parameterisations for numerical rainfall simulation in semi-humid and semi-arid catchments of Northern China, Atmos. Res., 191, 141–155, doi: 10.1016/j.atmosres.2017.03.012, 2017a.

Point 12: Section 2.2 can you provide more information of these catchments (e.g. catchment descriptors)? e.g. apart from catchment area, mean annual rainfall, catchment slope, mean flow, predominant land use and soil types, type of geology, percentage of urban area, etc. you can summarise all this info in a table. this is important to understand the catchment response to precipitation. Reply: Thanks for the reviewer's suggestion. Section 2.2, which changed as section 2.1 in the revised manuscript, has been revised as follows and the information table is also added to provide more details about the study area: "In this study, two mountainous sub-watersheds of the Daqing River basin (Fuping of the south branch and Zijingguan of the north branch) were chosen as the study area (Fig. 2 and Fig. 3). The Fuping sub-watershed has

a total area of 2,219 km2 and is located in the upper reaches of the Zhishahe River, a south branch of the Daqing River. The Zijingguan sub-watershed has a total area of 1,760 km2 and is located in the upper reaches of the Juma River, a north branch of the Daqing River. The two sub-watersheds are the most concentrated and typical cinnamon regions. The land use mainly includes grassland, farmland and forestland. The soil erosion is severe. Due to the dry soil conditions and the groundwater over-exploitation, the river has great seepage during the storm season. More information about the two sub-watersheds is shown in Table 1. The study area embodies the representative rainfall-runoff characteristics of the sub-humid and sub-arid area in Northern China. Rainfall in Northern China is characterized by summer storms with short durations and large intensities, which are likely to result in severe flood disasters especially in mountainous areas like Fuping and Zijingguan."

Point 13: Section 3.1.1. "in consistence" consistent? The grid resolution of the coarser domain is 9km, but the NCEP analysis is about 100km. How does the WRF model handles this discrepancy with the initial and boundary conditions between the outer domain and the analysis? What's the temporal resolution of the analysis? Reply: "consistent" is right and the sentence is revised accordingly. The WRF model is a next-generation mesoscale numerical weather prediction system. It can run on a variety of computing platforms and handle a broad range of applications across scales ranging from tens of meters to thousands of kilometers by dynamical downscaling. Actually when we run the WRF model, four nested domains are initially adopted, with the outermost domain being 27 km in order to deal with the discrepancy with the NCEP data. In order to avoid misunderstanding, the following sentences are added in Line 10-13, Page 5. In addition, the temporal resolution of the NCEP analysis data is 1 h. "The NCEP/NCAR final operational global analysis (FNL) data with spatial resolution of $1° \times 1°$ and temporal resolution of 1 h were used to provide the lateral and boundary conditions of the WRF model (Wang et al., 2013a,b). In order to eliminate the discrepancy of the initial and boundary conditions with the driven data, another outermost domain was set beyond the WRF three nested domains to downscale the FNL data to a spatial resolution of 27

km."

Point 14: Section 3.1.2. Need to define all acronyms for the physical parameterisations. Reply: Revised accordingly as below: "...which include two microphysics parameterisations, i,e, Purdue-Lin (Lin) (Lin et al., 1983) and WRF Single-Moment 6 (WSM6) (Hong et al., 2006), two cumulus parameterisations, i.e., Kain-Fritsch (KF) (Kain, 2004) and Grell-Devenyi (GD) (Grell and Freitas, 2014), and two PBL (planetary boundary layer) parameterisations, i.e., Mellor-Yamada-Janjic (MYJ) (Hong et al., 2006) and Yonsei University (YSU) (Janjić, 1994). Besides, Rapid Radiative Transfer Model (RRTM) and Dudhia (Evans et al., 2012) usually cooperated well as the long/short wave radiation parameterisations and Noah was chosen to be the land surface model (Chen et al., 2014)."

Point 15: Section 3.2.2 "... has widely been applied in Hebei Province ..." add references. Reply: The sentence is revised as below with the following reference added: "...has widely been applied in Hebei Province by considering both infiltration-excess and saturation-excess mechanisms of the runoff generation (Tian et al., 2019)." Reference: Tian, J., Liu, J., Yan, D., et al. Ensemble flood forecasting based on a coupled atmospheric-hydrological modeling system with data assimilation, Atmospheric Research, 224, 127-137, doi: 10.1016/j.atmosres.2019.03.029, 2019.

Point 16: How do you account for different soil types, land use, etc? These will have important implications in terms of runoff production. Do you use ET (evapotranspiration) or only E (evaporation)? do you have forest in any of the basins? Expand on the calibration and validation of the hydrological model, including both, lumped and distributed models. Reply: Thanks for the referee's remind. The type of soil in the study area is typical cinnamon soil with considerable soil erosions. The land use mainly includes grassland, farmland and forestland. Due to the perennial water shortage and groundwater overexploitation, the river has great seepage during the storm season. The Hebei model is specially developed as a very simple conceptual model for rainfall-runoff modeling in this region. As mentioned in section 3.2, the calculation of the lumped Hebei

model has no relation with the soil type or the land use, and the grid-based model only considers the spatial distribution of the rainfall, the soil water storage capacity and the soil infiltration capacity. As mentioned by Eqn. (4), E (evaporation) is used in the models instead of ET. The details of the study area are supplemented in a new table (Table 1). Please refer to our reply to Point 12.

Point 17: fig 11. Unclear if these results are for catchment A or B. Reply: As shown in Table 1, Event 1, 2 and 3 is from Fuping and Event 4 from Zijingguan. In order to make this clear, the captain of Fig. 11 is revised as follows: "Figure 11. Flood process simulations of the coupled atmospheric-hydrologic systems for the four storm events: (a) Event 1 in Fuping; (b) Event 2 in Fuping; (c) Event 3 in Fuping; (d) Event 4 in Zijingguan."

Point 18: Table 5. how do you isolate the impact of the different WRF microphysics parameterisations in your results? Reply: In this study, the impacts of different coupling scales are compared for each single storm event with the same parameterisations. The comparison is actually not carried out among the four events with different parameterisations. Please see our reply to Point 4 for more details.
* * *
**Two tables for the reply of Point 2**

The storm events used to calibrate the model

| Date | Sub-watershed | 24-h rainfall accumulations (mm) | Peak discharges (m³/s) |
|---|---|---|---|
| 02/07/1997 | Zijingguan | 51.31 | 163 |
| 05/07/1998 | Zijingguan | 68.37 | 129 |
| 29/06/2006 | Zijingguan | 48.55 | 100 |
| 03/07/2007 | Zijingguan | 36.96 | 25 |
| 01/09/2012 | Zijingguan | 38.58 | 20 |
| 11/08/2013 | Zijingguan | 40.74 | 33 |
| 06/07/2000 | Fuping | 60.86 | 330 |
| 24/07/2001 | Fuping | 56.03 | 105 |
| 13/08/2004 | Fuping | 32.85 | 102 |
| 14/08/2006 | Fuping | 24.64 | 41 |
| 30/08/2010 | Fuping | 20.43 | 33 |
| 29/06/2012 | Fuping | 25.42 | 48 |
| 29/06/2013 | Fuping | 19.68 | 38 |

The storm events used to verify the model

| Date | Sub-watershed | 24-h rainfall accumulations (mm) | Peak discharges (m³/s) |
|---|---|---|---|
| 06/06/1997 | Zijingguan | 57.44 | 128 |
| 30/06/2005 | Zijingguan | 46.22 | 98 |
| 14/08/2005 | Fuping | 34.92 | 103 |
| 24/07/2007 | Fuping | 42.16 | 106 |

**Fig. 1.**

Table 6 Calibrated parameters in the Hebei model

| Parameters | Units | Suggested values | Descriptions | parameter values for Fuping | parameter values for Zijingguan |
|---|---|---|---|---|---|
| $u$ | none | 0-0.1 | Decreasing speed of the infiltration rate with the increase of the soil moisture | 0.02 | 0.02 |
| $f_c$ | mm/h | 1-2 | stable infiltration rate | 1.5 | 1.5 |
| $n$ | none | 0.3-0.8 | exponent of the distribution curve for the infiltration capacity | 0.53 | 0.50 |
| $b$ | none | 0.3-0.5 | exponent of the distribution curve for the moisture storage capacity | 0.49 | 0.50 |
| $WMM$ | mm | 80-300 | maximal moisture storage capacity of a certain grid cell | 240 | 238 |
| $f_m$ | mm/h | 20-200 | maximum infiltration capacity of a certain grid cell | 120 | 120 |
| $A$ | $(m^3/s)^{-\omega}$ s | 0-1 | confluence parameter | 0.85 | 0.85 |

**Fig. 2.**

*Table 1. The characteristics of the two sub-watersheds*

| Descriptors | Fuping catchment | Zijingguan catchment |
|---|---|---|
| catchment area | 2,219 km$^2$ | 1,760 km$^2$ |
| mean annual rainfall | 490 mm | 650 mm |
| longitudinal river slope | 5.7% | 5.5% |
| Annual average runoff | 2.85×10$^8$ m$^3$ | 2.81×10$^8$ m$^3$ |
| predominant land use | grassland, farmland and forestland | grassland, farmland and forestland |
| soil type | cinnamon soil | cinnamon soil |
| type of geology | granitic gneiss | granitic gneiss |
| percentage of residential area | 0.63% | 0.52% |

**Fig. 3.**

---

## Author Comment (AC2) · 12 Apr 2020

Main comments: Point 1: More information of the Hebei model and the calibration of its parameters should be given. What is the advantage of using the Hebei model in the study area? What are the parameters calibrated in section 3.2 and how are they calibrated? Why choosing the 7 floods in Fuping and 6 floods in Zijingguan to calibrate the model? What are the values of the calibrated parameters finally used in the

coupled system? Reply: Thanks for the reviewer's suggestion. The Hebei rainfall-runoff model is specially developed to describe the runoff generation mechanisms in the semi-humid and semi-dry area of Northern China, which has been successfully applied in Hebei Province for rainfall-runoff modeling and real-time flood forecasting. Due to the perennial water shortage and groundwater overexploitation, both storage-excess and infiltration-excess is found with great seepage along the river channel during the storm season. The obvious advantage of the Hebei model is the consideration of both storage-excess and infiltration-excess mechanisms for rainfall-runoff generation. It is a well-known conceptual model in China, as popular as the Xin'anjinag model. The model is easily used, and can widely be applied to other semi-humid and semi-arid watersheds with complicated (both storage-excess and infiltration-excess) mechanisms for rainfall-runoff generation. The description for the storage-excess part in the Hebei model is the same as the Xin'anjiang model. On the other hand, to reflect the heterogeneity of the infiltration capacity across the catchment, a distribution curve is adopted and expressed as Eqn. (2). The Horton infiltration model is also applied to obtain the infiltration volume for the river channel seepage. When the calculation interval is one hour, the infiltration volume can be calculated by the Eqn. (16). In order to clarify this issue, the following sentences are added in Line 22-28, Page 6: "Due to the perennial water shortage and groundwater overexploitation, both storage-excess and infiltration-excess is found in the study area with great seepage along the river channel during the storm season. The obvious advantage of the Hebei model is the consideration of both storage-excess and infiltration-excess mechanisms for rainfall-runoff generation. The model is easily applied and can be used in other semi-humid and semi-arid watersheds. In the Hebei model, the description for the storage-excess part is the same as that in the Xin'anjiang model. On the other hand, the infiltration capacity across the watershed is described by a distribution curve described below, and the Horton model is applied to calculate the seepage along the river channel during the river routing." A new table below (Table 6) is added to show the calibrated parameter values on Page 6. The ranges of the parameter values are determined based on the

application experiences of the Hebei model, which has been used in Northern China for more than two decades. The SCE-UA (Shuffle Complex Evolution) method is used to calibrate the parameters of the Hebei model (Duan et al., 1994). Actually, we have very limited choices when selecting the calibration data. Considering the semi-humid and semi-dry conditions of the study area, the soil is relatively dry before the storm season, and there is not many storm events leading to significant peak discharges. In this case, 7 storm events in Fuping and 6 storm events in Zijingguan are chosen to calibrate the model. Detailed information (i.e., the cumulative rainfall amounts and the peak discharges) of the events are summarized in the table below. Considering there are already many table in the manuscript, this table is not shown. When calibrating the model, the calibration events are bounded together to calculate one NSE value as the objective function. In order to guarantee reasonable values for the initial model conditions, the 24-h storm event is not independently used, but with a continuous antecedent period of data with the length of 15-days before the start of the event. In this sense, the events used for calibrating the model is some kind of "continuous" time series data. The following sentences are added in Line 29, Page 10 and Line 1-7, Page 11 to supplement more details about the model calibration and validation: "The SCE-UA (Shuffle Complex Evolution) method (Duan et al., 1994) is used to calibrate the parameters and the calibrated values are shown in Table 6. Due to the limited observational data, 7 storm events in Fuping and 6 storm events in Zijingguan are selected and used to calibrate the Hebei model, and another 2 from each sub-watersheds are used for model validation. In order to guarantee reasonable values for the initial model conditions, the storm events are not independently used, but with an antecedent period of data with the length of 15-days before the start of the event. The validation results show an average NSE value of up to 0.686, indicating the calibrated models are reliable for further applications. It should be noted that the four storm events in Section 2.2 are different from those used for calibration and validation."

Reference: Duan, Q., Sorooshian, S., Gupta, V. K. Optimal use of the SCE-UA global optimization method for calibrating watershed models, J. Hydrol., 158(3-4), 265-284,

doi: 10.1016/0022-1694(94)90057-4, 1994.

Point 2: I think the gridded Hebei model is a semi-distributed model. The main goal of establishing the gridded Hebei model is to match the rainfall simulation from the NWP system. Hence, the Hebei model does not consider the spatial variability of the underlying condition of the watersheds. If so, I do not quite understand why the soil storage capacity and the infiltration capacity is discretized across the grid cells? Reply: Thanks for the referee's question. As the referee mentioned, the gridded Hebei model is a semi-distributed model and does not consider the spatial variability of the underlying condition of the watersheds. According to Eq. (4)-(6), the soil storage capacity and the infiltration capacity significantly affects and determines the runoff generation. In the lumped Hebei model, the two crucial elements are described by two distribution curves across the watershed (as shown by Fig. 7). When the gridded Hebei model is built, the soil storage capacity and the infiltration capacity needs to be determined in each grid cell. That is why the two elements are discretized in each of the grid cells. Based on the theory of the TOPMODEL, it can be assumed that areas with similar topographic indices have the same hydrological response. Experimentations carried out in the study area showed that the soil storage capacity and the infiltration capacity of different grid cells can be obtained and dispersed using the topographic indices as Eqn. (17) and (18).

Point 3: The errors of the coupled system generally come from two parts: the NWP system and the hydrologic model. Since the WRF model is used (the rainfall error of which is normally quite considerable), I believe the accuracy of the simulated rainfall is the main factor affecting the performance of the coupled system (although there might also be uncertainties from the hydrologic model). Could the authors specify the rainfall errors from each storm events and quantity how much the system errors come from the rainfall simulations? A further question is, how to improve the simulated rainfall from the NWP system in order to improve the performance of the coupled system. For example, some grid-based observations, such as QPEs from the weather radar

might be helpful. Reply: Thanks for the reviewer's suggestion. In the manuscript, Table 7 shows the simulation results of the coupled atmospheric-hydrologic systems based on WRF simulated rainfall for the four storm events, whereas Table 10 shows the simulation results of the coupled systems based on the corrected gridded rainfall for the four storm events. The differences of evaluation statistics between Table 7 and Table 10 reflects the system errors from the WRF rainfall simulations, which can be easily obtained by the subtraction of the corresponding values in Table 7 and 10. There are two main methods using the weather radar observations to improve the rainfall simulations, which are radar QPE or QPF and radar data assimilation for the NWP model. The following paragraph is added in Line 25-32, Page 13 and Line 1-4, Page 14, and a new table (Table 11) is further added. "Comparing the results from Table 7 and Table 10, the system errors from the rainfall simulations (as shown in Table 11) can be easily obtained by the subtraction of the corresponding values in Table 7 and Table 10. For event 1, the average |Rl|-|Rl-corrected|, |Rf|-|Rf-corrected| and NSE-corrected-NSE of the three different grid sizes caused by the rainfall simulations is 7.26%, 7.00% and 0.1469. In the same way, the average |Rl|-|Rl-corrected|, |Rf|-|Rf-corrected| and NSE-corrected-NSE of the three grid sizes is 7.47%, 6.34% and 0.1116 for event 2. A notable case is event 3. |Rl|-|Rl-corrected| of event 3 with the grid size 3×3 km (7.96%) is the highest among the three grid sizes, and the highest |Rf|-|Rf-corrected| (3.56%) comes from the grid size 9×9 km. Due to the errors of the rainfall simulations, all the NSEs decline more than 0.5 for the three grid sizes. For event 4, the average |Rl|-|Rl-corrected|, |Rf|-|Rf-corrected| and NSE-corrected-NSE of the three grid sizes caused by the rainfall simulations is 7.32%, 6.58% and 0.0991. It can easily be found that the magnitudes of most errors in Table 11 are higher than those of Table 10, which indicates that the accuracy of the simulated rainfall is the main factor affecting the performance of the coupled system. In order to improve the rainfall simulation in small and medium scale catchments, radar data with high spatiotemporal resolution should be a good choice, such as radar QPE or QPF and radar data assimilation for the NWP model (Xiao and Sun, 2007; Harader et al., 2012)."

References: Xiao, Q., Sun, J. Multiple-Radar Data Assimilation and Short-Range Quantitative Precipitation Forecasting of a Squall Line Observed during IHOP_2002, Mon. Weather Rev., 135(10), 3381-3404. doi: 10.1175/MWR3471.1, 2007. Harader, E., Borrell-Estupina, V., Ricci, S., et al. Correcting the radar rainfall forcing of a hydrological model with data assimilation: application to flood forecasting in the Lez catchment in Southern France, Hydrol. Earth Syst. Sci., 16, 4247–4264, doi: 10.5194/hess-16-4247-2012, 2012.

Point 4: I agree that Cv is used to describe the evenness of rainfall for both spatial and temporal distributions. However, a critical value of 0.40 for evenness in space and 1.00 for evenness in time, is hard to follow. Explain how the threshold is obtained. You can say event 1 has relatively even distributed rainfall according to Cv rather than using the value of 0.4 as a threshold. Reply: Thanks for the reviewer's suggestion. The spatial and temporal Cv values of the historical storm events from 1985 to 2018 in the study area are calculated to analyse the characteristics of the rainfall evenness. In reality, in comparison to the southern part of China, it is difficult to find absolute even rainfall events in Northern China in either spatial or temporal dimensions. In this study, a threshold of 5% is used to separate even and uneven storms. Thus, the storm events with a spatial Cv < 0.4 or with a temporal Cv < 1.0 can both account for 5% of the total storm events from 1985 to 2018. It should be mentioned that values of 0.4 and 1.0 are calculated by statistical analyses of the historical storm events, thus may not be transferable to other areas with different meteorological conditions. In the revised manuscript, the related descriptions for these two thresholds are removed. Instead, the spatial and temporal evenness of rainfall distribution is ranked among different storm events. The following sentences can be found in Line 22-24 Page 4: "The smaller is the value of Cv, the more even is the rainfall distribution in space or time. According to Table 3, the ranking of the distribution evenness of rainfall in space is event 2 > event 1 > event 4 > event 3 and that in time is event 1 > event 2 > event 4 > event 3."

Point 5: It is concluded from the study that for storm events with uneven rainfall distributions, a finer coupling scale can lead to a better performance of the coupled system, however, the coupling scale shows less impact on the system for events with uneven distributions. To my opinion, these conclusions are highly dependent on the case studies. Considering the study only focusing on two semi-humid and semi-dry watersheds with limited storm events involved, it is better to point out that the results are some kind of site-specific. More case studies are needed before more general conclusions can be achieved. Reply: We are grateful for the reviewer's kindly remind. The following sentences are added in Line 21-23 Page 13: "Considering the study only focusing on two semi-humid and semi-dry watersheds with limited storm events involved, the results in this study should be verified by more case studies before more general conclusions can be achieved."

Spelling and grammar mistakes should be checked carefully throughout the manuscript: Page 2, line 5 and line 9: "atmosphere-hydrologic" should be "atmospheric-hydrologic" Page 5, line 11: "Based on the historical storm events in the study area, and using 5% as a cutoff" should be "Based on the historical storm events in the study area by using 5% as a cutoff" Page 10, line 21: "Grid center coordinates...for driving the hydrologic model" should be "The coordinates of the grid cell centers...to drive the hydrologic model" Page 11, line 22: "...three grid sizes led to different simulation results for different rainfall events" should be "...different rainfall events have different simulation results with the three grid sizes". Page 12, line 4: "Considering the spatial distribution characteristics of the rainfall..." should be "Considering the characteristics of the spatial rainfall distributions..." Page 12, line 29: "...the WRF model had the ability to reflect the spatial distribution of the rainfall" should be "...the WRF model was able to capture the spatial patterns of the simulated rainfall..." Page 13, line 4 "...similar simulation results with three different grid sizes..." should be "...similar simulation results of the three different grid sizes..." Reply: All the spelling and grammar mistakes are revised accordingly. The whole manuscript is checked through carefully with other typos corrected.
The storm events used to calibrate the model

| Date | Sub-watershed | 24-h rainfall accumulations (mm) | Peak discharges ($m^3/s$) |
|---|---|---|---|
| 02/07/1997 | Zijingguan | 51.31 | 163 |
| 05/07/1998 | Zijingguan | 68.37 | 129 |
| 29/06/2006 | Zijingguan | 48.55 | 100 |
| 03/07/2007 | Zijingguan | 36.96 | 25 |
| 01/09/2012 | Zijingguan | 38.58 | 20 |
| 11/08/2013 | Zijingguan | 40.74 | 33 |
| 06/07/2000 | Fuping | 60.86 | 330 |
| 24/07/2001 | Fuping | 56.03 | 105 |
| 13/08/2004 | Fuping | 32.85 | 102 |
| 14/08/2006 | Fuping | 24.64 | 41 |
| 30/08/2010 | Fuping | 20.43 | 33 |
| 29/06/2012 | Fuping | 25.42 | 48 |
| 29/06/2013 | Fuping | 19.68 | 38 |

**Fig. 1.**

Table 6 Calibrated parameters in the Hebei model

| Parameters | Units | Suggested values | Descriptions | parameter values for Fuping | parameter values for Zijingguan |
|---|---|---|---|---|---|
| $u$ | none | 0-0.1 | Decreasing speed of the infiltration rate with the increase of the soil moisture | 0.02 | 0.02 |
| $f_c$ | mm/h | 1-2 | stable infiltration rate | 1.5 | 1.5 |
| $n$ | none | 0.3-0.8 | exponent of the distribution curve for the infiltration capacity | 0.53 | 0.50 |
| $b$ | none | 0.3-0.5 | exponent of the distribution curve for the moisture storage capacity | 0.49 | 0.50 |
| $WMM$ | mm | 80-300 | maximal moisture storage capacity of a certain grid cell | 240 | 238 |
| $f_m$ | mm/h | 20-200 | maximum infiltration capacity of a certain grid cell | 120 | 120 |
| $A$ | $(m^3/s)^{-\omega}s$ | 0-1 | confluence parameter | 0.85 | 0.85 |

**Fig. 2.**

Table 11. The system errors from the rainfall simulations for four storm events.

| Storm event | Grid size | $|R_l|-|R_{l\text{-}corrected}|$ (%) | $|R_f|-|R_{f\text{-}corrected}|$ (%) | $NSE_{\text{-}corrected}$–$NSE$ |
|---|---|---|---|---|
| | 1×1 km | 9.39 | 7.06 | 0.0831 |
| Event 1 | 3×3 km | 7.02 | 7.84 | 0.2790 |
| | 9×9 km | 5.36 | 6.11 | 0.0785 |
| | 1×1 km | 6.57 | 5.77 | 0.0964 |
| Event 2 | 3×3 km | 5.96 | 6.40 | 0.0884 |
| | 9×9 km | 9.89 | 6.86 | 0.1499 |
| | 1×1 km | 2.28 | 1.08 | 0.5357 |
| Event 3 | 3×3 km | 7.96 | 2.07 | 0.5659 |
| | 9×9 km | 1.19 | 3.56 | 0.5738 |
| | 1×1 km | 5.62 | 5.89 | 0.0301 |
| Event 4 | 3×3 km | 7.13 | 6.88 | 0.0660 |
| | 9×9 km | 9.20 | 6.96 | 0.2011 |

**Fig. 3.**

---

## Author Response (AR2)

**A coupled atmospheric-hydrologic modeling system with variable grid sizes for rainfall-runoff simulation in semi-humid and semi-arid watersheds: How does the coupling scale affects the results?**

**Comments:**

**Point 1:** Page 5, I do not think the authors can 'eliminate the modeling errors caused by choosing inappropriate WRF parameterizations', please avoid the unreliable conclusion.

**Reply:** Thanks for the reviewer's suggestion. The unreliable conclusion is removed and the sentence is revised as:

*"The most suitable physical parameterisations resulting the best rainfall simulations (Tian et al., 2017a) are used for each of the four storm events, as shown in Table 5."*

**Point 2:** Section 3.2, there are only few references in describing the proposed hydrological model. I would suggest the authors to put more references to support some key concepts.

**Reply:** According to the reviewer's suggestion, three references are added in Section 3.2.

Reference:

Zhao, R. The Xinanjiang model applied in China, J. Hydrol., 135(1-4), 371-381, doi: 10.1016/0022-1694(92)90096-E, 1992.

Horton, R. E. The role of infiltration in the hydrologic cycle, Trans. A. G. U., 14(1), 446-460, doi: 10.1029/TR014i001p00446, 1933.

Goutal, N., Sainte-Marie, J. A kinetic interpretation of the section-averaged Saint-Venant system for natural river hydraulics, Int. J. Numer. Meth. Fl., 67(7): 914-938, doi: 10.1002/fld.2401, 2011.

**Point 3:** Page 14, please explain the abbreviation of QPE and QPF.

**Reply:** Thanks for the reviewer's suggestion. The sentence is revised as:

*"...such as radar Quantitative Precipitation Estimates (QPE) or Quantitative Precipitation Forecasts (QPF) and radar data assimilation for the NWP model..."*